

# Groundwater dynamics beneath a marine ice sheet

Gabriel J. Cairns[1], Graham P. Benham[1,2], and Ian J. Hewitt[1]

[1]Mathematical Institute, University of Oxford, Woodstock Road, OX2 6GG, UK
[2]School of Mathematics and Statistics, University College Dublin, Belfield, Dublin 4, Ireland

**Correspondence:** Gabriel Cairns (cairns@maths.ox.ac.uk)

**Abstract.** Sedimentary basins beneath many Antarctic ice streams host substantial volumes of groundwater, which can be exchanged with a "shallow" subglacial hydrological system of till and channelised water. This exchange contributes substantially to basal water budgets, which in turn modulate the flow of ice streams. The geometry of these sedimentary basins is known to be complex, and the groundwater therein has been observed to vary in salinity due to historic seawater intrusion. However, little is known about the hydraulic properties of subglacial sedimentary basins, and the factors controlling groundwater exfiltration and infiltration. We develop a mathematical model for two-dimensional groundwater flow beneath a marine-terminating ice stream on geological timescales, taking into account the effect of seawater intrusion. We find that seawater may become "trapped" in subglacial sedimentary basins, through cycles of grounding line advance and retreat or through "pockets" arising from basin geometry. In addition, we estimate the sedimentary basin permeability which reproduces field observations of groundwater salinity profiles from beneath Whillans Ice Stream in West Antarctica. Exchange of groundwater with the shallow hydrological system is primarily controlled by basin geometry, with groundwater being exfiltrated where the basin becomes shallower and re-infiltrating where it becomes deeper. However, seawater intrusion also has non-negligible effects on this exchange.

## 1 Introduction

In order to understand the stability of the West Antarctic Ice Sheet and its potential contribution to future sea level rise, it is critical to understand the dynamics of marine-terminating glaciers (Smith et al., 2020). The flow of such glaciers is strongly modulated by a shallow hydrological system at the base of the ice, which includes pressurised water in films, channels and lakes, along with a water-saturated layer of deformable sediments (Tulaczyk et al., 2000; Bougamont et al., 2011; Gustafson et al., 2022). In addition, the fast-flowing glaciers known as ice streams which drain much of West Antarctica typically overlie large sedimentary basins, which host substantial volumes of groundwater. The exchange of this deeper groundwater with the shallow hydrological system has emerged as an important contributor to the basal conditions which dominate ice stream flow (Christoffersen et al., 2014; Siegert et al., 2018).

Radar measurements of the sedimentary basin beneath the Ross Ice Shelf have revealed a complex geometry, consisting of peaks, troughs and faults (Tankersley et al., 2022). Moreover, magnetotelluric (MT) measurements of electrical conductivity have shown an increase in salinity with depth in these aquifers, which has been attributed to seawater intrusion during a Holocene glacial minimum (Gustafson et al., 2022).





Such evidence of past seawater intrusions is useful for constraining the grounding line history of the West Antarctic ice sheet, and thus gaining insight into its future stability (Kingslake et al., 2018; Venturelli et al., 2020, 2023). Previous studies have used evidence of historic seawater intrusion to infer the effects of continental ice sheets on groundwater systems during glaciations (Aquilina et al., 2015).

Several recent studies have considered the mathematical modelling of groundwater in subglacial sedimentary basins. These studies have focused on exchange of groundwater with the shallow hydrological system, driven by the poro-elastic response of sedimentary basins to ice overpressure changes on timescales of decades to centuries (Gooch et al., 2016; Li et al., 2022; Robel et al., 2023). However, these models mostly focus on vertical flow (although Li et al. (2022) consider two-dimensional flow), and do not consider the effect of salinity differences or sedimentary basin geometry.

In contrast, the mathematical theory of seawater intrusion in coastal aquifers is of longstanding interest due to its implications for groundwater management, beginning with the investigations of Badon-Ghyben and Herzberg (1896). A rich array of mathematical and numerical techniques have emerged to investigate the effects of overextraction and, more recently, sea level rise on coastal groundwater flow (Bear, 2013; Werner et al., 2013; Ketabchi et al., 2016; Mondal et al., 2019; Richardson et al., 2024). However, such models are yet to be applied to a subglacial context.

In this paper, we consider a mathematical model for groundwater flow in a sedimentary basin beneath a marine ice sheet on geological timescales, accounting for seawater intrusion, ice sheet growth and retreat, and basin geometry. We show that grounding line motion and basin geometry may give rise to "saltwater trapping", where seawater remains resident in portions of the aquifer that would be saturated with freshwater in the simplest steady state. The phenomenon of saltwater trapping depends heavily on hydraulic properties of the aquifer, which are summarised in a dimensionless parameter $K$, along with past 45 grounding line motion.

We then apply this model to the sedimentary basin beneath the Ross Ice Shelf in West Antarctica, enabling us to predict how groundwater is exchanged with the shallow hydrological system. In addition, comparing modelled saltwater trapping to field observations enables us to predict appropriate values of the parameter $K$. Our results evidence the importance of accounting for horizontal flow and saltwater intrusion when modelling groundwater dynamics in a subglacial sedimentary basin, and the 50 wider implications of these sedimentary basins for subglacial hydrology.

## 2 Model design

### 2.1 Derivation of the governing equation

We investigate seawater intrusion beneath a marine ice sheet in a vertical cross-section of an aquifer, given by $0 < x < x_g(t)$, $b(x) < z < S(x)$. The setup is depicted in Figure 1. Here $x_g$ represents the grounding line, where the grounded ice sheet 55 transitions to a floating ice shelf, and $x = 0$ represents the water divide within the aquifer. For $x > x_g$, the upper surface of the aquifer is therefore exposed to the ocean, which rapidly saturates this region of the aquifer with seawater. In addition, the floating ice shelf necessarily imposes the same overpressure as the seawater it displaces. We take the present-day sea level as the reference $z = 0$.





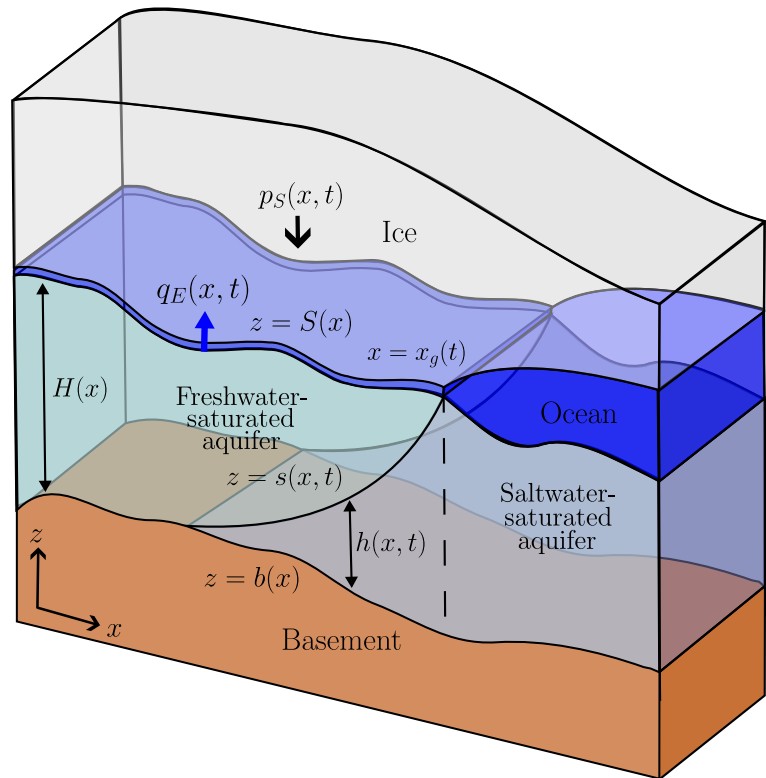

**Figure 1.** Schematic of model. The aquifer is bounded by $z = b(x)$ below and $z = S(x)$ above, and has thickness $H(x) = S(x) - b(x)$. The saltwater region occupies $b(x) < z < s(x)$ and has thickness $h(x) = s(x) - b(x)$, and the freshwater region occupies $s(x) < z < S(x)$ and has thickness $H(x) - h(x)$. The "shallow" subglacial hydrological system is depicted as a layer of water between the ice and the aquifer at $z = S$. The grounding line $x = x_g(t)$ is marked by a vertical dashed line.

The surfaces $z = b(x)$ and $z = S(x)$ represent the basement and upper surface of the aquifer, which therefore has a thickness

$H(x) = S(x) - b(x)$. We assume that the aquifer is homogeneous, with porosity $\phi$ and permeability $k$, although a generalisation of the model to relax these assumptions is straightforward (as discussed in Appendix B). For $x < x_g(t)$, the upper surface $z = S(x)$ is overlain by an ice sheet above a lubricating layer of freshwater. The aquifer is assumed to be saturated at all times, but exchanges water with this lubricating layer at a rate $q_E(x, t)$, where $q_E > 0$ corresponds to exfiltration of water from the sedimentary basin. This layer represents an idealised basal hydrological system.

We also assume that the aquifer is rigid. This assumption is valid provided that any dynamic deformation of the sediments is small. For a rigid aquifer, $q_E$ represents the rate of exfiltration driven by horizontal pressure gradients emerging from topography, ice thickness and seawater intrusion on long timescales. However, in reality the poro-elastic response of the sediments produces an additional exfiltration over shorter timescales, which has been considered in previous models by Li et al. (2022) and Robel et al. (2023). These models find that this exfiltration may reach tens to hundreds of millimetres per year for a





highly permeable aquifer. This distinct exfiltration, which we do not consider, requires further modelling to combine with the long-timescale exfiltration $q_E(x,t)$.

We use the sharp-interface approximation, a frequent assumption in saltwater intrusion problems, which asserts that mixing between freshwater and saltwater through molecular diffusion and hydraulic dispersion is negligible (Bear, 2013; Mondal et al., 2019). The aquifer therefore consists of a freshwater-saturated region $s(x,t) < z < S(x)$ and a saltwater-saturated region

$b(x) < z < s(x,t)$, where $z = s(x,t)$ is the position of the "sharp" freshwater-saltwater interface. Freshwater and saltwater are physically distinguished by their densities $\rho_f$ and $\rho_s$. We use $h(x,t) = s(x,t) - b(x)$ to denote the thickness of the saltwater-saturated region, so that the thickness of the freshwater-saturated region is $H(x) - h(x,t)$. The following model is solved for $h(x,t)$, along with the pore pressure $p(x,z,t)$ and the Darcy flux $(u(x,z,t), w(x,z,t))$ of water in the aquifer.

The model consists of mass conservation for the pore water,

$$\frac{\partial u}{\partial x} + \frac{\partial w}{\partial z} = 0, \tag{1}$$

along with the horizontal component of Darcy's law:

$$u = -\frac{k}{\mu}\frac{\partial p}{\partial x}. \tag{2}$$

Here $k$ is the permeability of the porous medium, assumed constant for simplicity, and $\mu$ is the fluid viscosity, which we assume is the same for fresh and salty water. These equations are combined with the Dupuit approximation, namely that the pressure

is hydrostatic:

$$\frac{\partial p}{\partial z} = -\rho_j g, \qquad j \in \{s, f\}. \tag{3}$$

This approximation is standard in models of groundwater flow and is justified provided that the aspect ratio of the aquifer is small, i.e. the aquifer is long and thin, which is true for our purposes (see Tables 1 and 2) (Gustafson et al., 2022). Equation (3) can equivalently be found as the vertical component of Darcy's law in a shallow limit.

We next prescribe boundary conditions for the model. The basement $z = b(x)$ is taken to be rigid and impermeable:

$$u\frac{\partial b}{\partial x} - w = 0 \quad \text{on} \quad z = b(x). \tag{4}$$

In addition, the sharp interface is taken to be kinematic,

$$\phi\frac{\partial h}{\partial t} + u_\pm\frac{\partial s}{\partial x} - w_\pm = 0 \quad \text{on} \quad z = s(x,t), \tag{5}$$

where the subscripts $\pm$ reflect that the Darcy flux components are discontinuous across this interface. However, the pressure

$p(x,z,t)$ must be continuous.

On the upper surface, we prescribe that the pressure of the pore fluid is continuous with that of the lubricating layer, and moreover assume that this pressure is equal to the hydrostatic ice overburden pressure:

$$p = p_S(x,t) = \rho_i g H_i(x,t) \quad \text{on} \quad z = S(x), \tag{6}$$





where $\rho_i$ and $H_i(x,t)$ are the density and thickness of ice respectively. This boundary condition is standard in models of
subglacial aquifers, and is equivalent to stipulating that the "effective pressure" is zero (Lemieux et al., 2008; Gooch et al., 2016; Li et al., 2022; Robel et al., 2023).

The Darcy flux of water exfiltrated by the aquifer is then given by

$$q_E = \left( w - u \frac{\partial S}{\partial x} \right)\Bigg|_{z=S(x)}, \tag{7}$$

where $q_E < 0$ corresponds with the aquifer being recharged by basal water from the lubricating layer.

In addition, horizontal boundary conditions are required. At the water divide $x=0$, we impose that there is no horizontal flux of water:

$$Hu = 0 \quad \text{at} \quad x = 0. \tag{8}$$

At the grounding line $x = x_g(t)$, the weight of the ice sheet is that of the displaced ocean, and the aquifer is saturated with seawater. The pressure is therefore hydrostatic:

$$p = -\rho_s g z \quad \text{at} \quad x = x_g, \tag{9}$$

taking the atmospheric pressure as the reference pressure. Note that by definition

$$p_S = \rho_i g H_i = -\rho_s g S \quad \text{at} \quad x = x_g, \tag{10}$$

since this is where the ice achieves flotation.

We now seek to reduce the above model to an evolution equation for $h(x,t)$. Firstly, integrating (3) subject to the condition
(6) gives the pressure as

$$p = \begin{cases} p_S + \rho_f g(S-z) & \text{for} \quad s < z < S, \\ p_S + \rho_f g\left(S - z + \delta(s-z)\right) & \text{for} \quad b < z < s, \end{cases} \tag{11}$$

where

$$\delta = \frac{\rho_s - \rho_f}{\rho_f} \approx 0.025, \tag{12}$$

is the scaled density difference between fresh and saltwater. Darcy's law (2) then gives the horizontal Darcy flux:

$$u = \begin{cases} -\frac{k\rho_f g}{\mu} \frac{\partial}{\partial x}\left(\frac{p_S}{\rho_f g} + S\right) & \text{for} \quad s < z < S, \\ -\frac{k\rho_f g}{\mu} \frac{\partial}{\partial x}\left(\frac{p_S}{\rho_f g} + S + \delta s\right) & \text{for} \quad b < z < s. \end{cases} \tag{13}$$

Note that in order to be consistent with the condition (8), we need that

$$H \frac{\partial}{\partial x}\left(\frac{p_S}{\rho_f g} + S\right) = 0 \quad \text{at} \quad x = 0. \tag{14}$$





We now take the mass conservation statement Eq. (1) and integrate over the saltwater region $b(x) < z < s(x,t)$, subject to the kinematic conditions (4) and (5), yielding a statement of depth-integrated mass conservation

$$\phi\frac{\partial h}{\partial t} + \frac{\partial}{\partial x}\int_b^s u\,\mathrm{d}z = 0. \tag{15}$$


Substituting in (13) gives the desired equation for $h(x,t)$:

$$\phi\frac{\partial h}{\partial t} = \frac{k\rho_f g}{\mu}\frac{\partial}{\partial x}\left[h\frac{\partial}{\partial x}\left(\frac{p_S}{\rho_f g} + S + \delta s\right)\right], \tag{16}$$

recalling that $s = b + h$. This equation requires two boundary conditions in $x$. The no-flux condition (8) implies that, as well as (14), we need

$$h\frac{\partial}{\partial x}\left(\frac{p_S}{\rho_f g} + S + \delta s\right) = 0 \quad \text{at} \quad x = 0. \tag{17}$$


In addition, the condition (9) gives

$$h = H \quad \text{at} \quad x = x_g. \tag{18}$$

A similar depth integration of Eq. (1) over the entirety of the aquifer $b < z < S$, using the definition (7), provides an equation for the exfiltration $q_E$:

$$q_E = -\frac{k\rho_f g}{\mu}\frac{\partial}{\partial x}\left[H\frac{\partial}{\partial x}\left(\frac{p_S}{\rho_f g} + S\right) + \delta h\frac{\partial s}{\partial x}\right]. \tag{19}$$


The first term, containing $H$, depends on the basement geometry and ice overpressure, and therefore depends on time only through the instantaneous position of the ice sheet. The second, containing $h$, accounts for the effect of saltwater intrusion on the freshwater recharge or discharge.

## 2.2 Non-dimensionalisation

We assume that the grounding line position $x_g(t)$ and aquifer thickness $H(x,t)$ provide suitable horizontal and vertical length-scales $[x]$ and $[z]$. We also assume that grounding line motion imposes a natural timescale $[t]$. We then scale


$$x, x_g \sim [x], \quad z, h, H, b, s, S \sim [z], \quad t \sim [t], \quad p_S \sim \rho_f g[z]. \tag{20}$$

Under these scalings, Eq. (16) becomes

$$\frac{\partial h}{\partial t} = K\frac{\partial}{\partial x}\left[h\frac{\partial}{\partial x}(p_S + S + \delta s)\right], \tag{21}$$

where

$$K = \frac{k\rho_f g[z][t]}{\phi\mu[x]^2}, \tag{22}$$



**Table 1.** Dimensional scalings used in numerical solutions.

| Scaling | Value (Range) | Reference(s) |
|---|---|---|
| Vertical lengthscale $[z]$ | $1 \times 10^3$ m | van der Wel et al. (2013); Gustafson et al. (2022); Li et al. (2022) |
| Horizontal lengthscale $[x]$ | $5 \times 10^5$ m | Peters et al. (2006); Gustafson et al. (2022); Li et al. (2022) |
| Gravitational acceleration $g$ | 9.81 m s$^{-2}$ | |
| Viscosity $\mu$ | $1 \times 10^{-3}$ Pa s | Bear (2013) |
| Permeability $k$ | $10^{-12}$ ($10^{-15}$–$10^{-12}$) m$^2$ | Gooch et al. (2016); Li et al. (2022); Robel et al. (2023) |
| Freshwater density $\rho_f$ | $1.0 \times 10^3$ kg m$^{-3}$ | Bear (2013) |
| Saltwater density $\rho_s$ | $1.025 \times 10^3$ kg m$^{-3}$ | Bear (2013) |
| Porosity $\phi$ | 0.43 (0.1–0.6) | Gooch et al. (2016); Michaud et al. (2017); Gustafson et al. (2022) |
| Timescale $[t]$ | 100 kyr | Pollard and DeConto (2009); Willeit et al. (2019) |
| Glen's law parameter $A$ | $2 \times 10^{-23}$ kg$^{-3}$ m$^3$ s$^5$ (Sect. 5) | Cuffey and Paterson (2010) |
| Accumulation $a$ | 0.1 m yr$^{-1}$ (Sect. 5) | Cuffey and Paterson (2010) |

**Table 2.** Values of dimensionless parameters used in numerical solutions.

| Parameter | Value (Range) | Definition |
|---|---|---|
| Aspect ratio $\varepsilon$ | $2 \times 10^{-3}$ | $\varepsilon = [z]/[x]$ |
| Density difference $\delta$ | 0.025 | $\delta = (\rho_s - \rho_f)/\rho_f$ |
| Dimensionless hydraulic conductivity $K$ | 0.0035–3.5 | Eq. (22) |
| Dimensionless accumulation $\alpha$ | 5 (Sect. 5) | Eq. (30) |

is the dimensionless hydraulic conductivity of the groundwater flow. A larger $K$ corresponds to faster groundwater flow, and vice versa. The conditions (17) and (18) become

$$h\frac{\partial}{\partial x}\left(p_S + S + \delta s\right) = 0 \quad \text{at} \quad x = 0, \tag{23}$$


$$h = H \quad \text{at} \quad x = x_g. \tag{24}$$

We can also obtain from Eq. (19) a dimensionless expression for the exfiltration flux, scaled with $[z]/[t]$:

$$q_E = K\frac{\partial}{\partial x}\left[H\frac{\partial}{\partial x}\left(p_S + S\right) + \delta h\frac{\partial s}{\partial x}\right]. \tag{25}$$

Values of the scalings in Eq. (20), and the corresponding values of dimensionless parameters, are provided in Tables 1 and 2

respectively.

In deriving Eq. (21), we have assumed that the top of the saltwater region $z = s(x,t)$ is a kinematic interface. However, it is possible for the aquifer to become saturated with saltwater throughout its depth, so that $s(x,t) = S(x)$ (i.e. $h = H$). If this is





the case, then saltwater, rather than freshwater, is discharged across the upper surface at a rate

$$q_E = K \frac{\partial}{\partial x} \left[ H \frac{\partial}{\partial x} \left( p_S + S + \delta \frac{\partial S}{\partial x} \right) \right]. \tag{26}$$

The state in which $s = S$ can be maintained as long as the outflux given by Eq. (26) is non-negative; otherwise, freshwater influx will create a freshwater layer.

To allow for this case, we replace Eq. (21) by the following complementarity problem:

$$(H - h) \left( -\frac{\partial h}{\partial t} + K \frac{\partial}{\partial x} \left[ h \frac{\partial}{\partial x} (p_S + S + \delta s) \right] \right) = 0,$$

$$-\frac{\partial h}{\partial t} + K \frac{\partial}{\partial x} \left[ h \frac{\partial}{\partial x} (p_S + S + \delta s) \right] \geq 0, \tag{27}$$

$$H - h \geq 0.$$

This formulation establishes that either Eq. (21) applies and $h \leq H$, or $h = H$ and saltwater is discharged according to (26).
We describe the numerical method used to solve this problem in Appendix A.

In Sect. 3 we consider the solution of Eq. (27) in a uniform aquifer geometry, firstly looking at steady-state solutions, before moving to a periodic time-dependent solution, which introduces the possibility of saltwater trapping via grounding line movement. In Sect. 4, we generalise to non-uniform aquifer geometries, which enables further potential saltwater trapping via so-called pockets. Finally in Sect. 5, we use Eq. (27) to model groundwater flow in the sedimentary basin beneath the Ross Ice
Shelf.

## 2.3    Ice sheet forcing

The ice overpressure $p_S$ is related to the ice thickness $H_i$ by Eq. (6). A classical ice sheet model uses a mass and momentum balance, combined with a condition such as Eq. (10) at the grounding line, to determine the evolution of the ice sheet thickness together with the grounding line position (Schoof, 2007; Cuffey and Paterson, 2010).
In our case, we use the shallow ice approximation, assuming negligible bed slip, with a Glen's law rheology (with exponent $n = 3$), and a uniform accumulation rate. We also assume for simplicity that the dynamics of the ice sheet are quasi-steady for a given $x_g$. This leads to the dimensional equation

$$\frac{2}{5} A (\rho_i g)^3 H_i^5 \left| \frac{\partial}{\partial x} (H_i + S) \right|^3 = ax, \tag{28}$$

where $A$ is a rheological constant and $a$ is the constant accumulation rate (Cuffey and Paterson, 2010). In dimensionless
variables, after scaling $H_i$ with $[z]$, this becomes

$$p_S = r_i H_i, \quad H_i^5 \left| \frac{\partial}{\partial x} (H_i + S) \right|^3 = \alpha x \tag{29}$$

where

$$r_i = \frac{\rho_i}{\rho_f} \approx 0.917, \quad \alpha = \frac{5a[x]^4}{2A (\rho_i g)^3 [z]^8}. \tag{30}$$





Equation (29) is consistent with the condition (14) provided that $\partial S/\partial x(0) = 0$, and is solved by numerical integration subject
to the condition (10). In dimensionless terms, Eq. (10) becomes

$$H_i = -\frac{1+\delta}{r_i}S \quad \text{at} \quad x = x_g, \tag{31}$$

Since $r_i$ is known, this model effectively contains one parameter $\alpha$, which we treat as constant (although another version
of this model could have $\alpha$ varying with time to reflect changes in accumulation). Sensitivity of $p_S$ to $\alpha$ is low due to the
high powers appearing in Eq. (29). Our chosen value of $\alpha$ in Sect. 5 is based on present-day measurements and historic
reconstructions of ice thickness, and is consistent with realistic values of $A$ and $a$. In Sect. 3 and Sect. 4, a lower but plausible
value of $\alpha$ is chosen for illustrative purposes.

## 3 Solutions in an idealised aquifer geometry

### 3.1 Steady states

Although, for realistic parameter values (see Table 1), the system is usually far from a steady state, it is useful to understand
these states in order to gain insight into solutions of Eq. (27) more generally. The steady states of (21) satisfy

$$h\frac{\partial}{\partial x}\left(p_S + S + \delta s\right) = 0, \tag{32}$$

using the no-flux condition (23). We therefore have

$$h = 0 \quad \text{or} \quad p_S + S + \delta s = \text{const}. \tag{33}$$

By the condition (24), the latter must hold near the grounding line $x_g$. There are therefore two basic forms of steady-state
solution (excluding possible steady states of the complementarity problem (27) in which $s = S$ over some region), a "lens"
case and "nose" case (Bear, 2013), which we discuss below.

### 3.1.1 Lens case

One possibility is that $h > 0$ everywhere, resulting in a freshwater "lens" beneath the ice sheet. In this case, conditions (10)
and (24) imply

$$p_S + S + \delta s = 0 \quad \text{for} \quad 0 < x < x_g. \tag{34}$$

Since we must have $s > b$, such a solution requires that

$$p_S + S + \delta b < 0 \quad \text{for} \quad 0 < x < x_g. \tag{35}$$

This generally occurs when the ice sheet is shallow relative to the depth of the sedimentary basin, or when the grounding line
has retreated far inland. The interface in the lens solution does not coincide with the upper surface $z = S$, provided that $s < S$,
i.e.

$$S + \delta b \leq -p_S \leq (1+\delta)S. \tag{36}$$



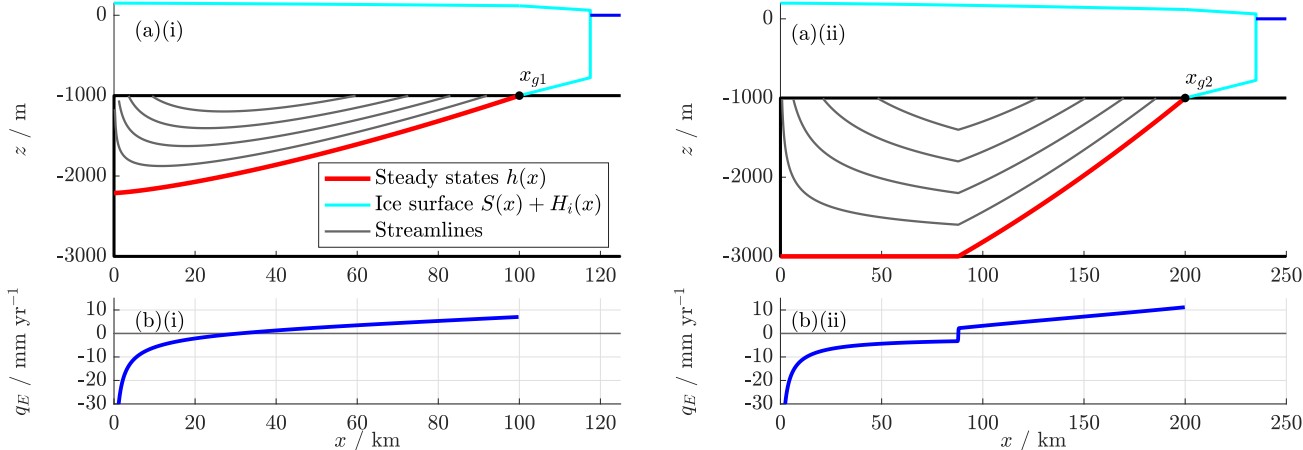

**Figure 2.** (a) Interface $s(x)$ and (b) exfiltration flux $q_E$, for (i) "lens" and (ii) "nose" steady states arising from different grounding line positions $x_{g1}$ and $x_{g2}$, for a uniform depth aquifer with $b = -3000$ m, $S = -1000$ m. Parameters are as listed in Tables 1 and 2, with $k = 10^{-12}$ m$^2$ ($K = 0.41$) and $\alpha = 0.1$.

### 3.1.2 Nose case

If the condition (35) is violated for some $x$, then the lens solution must be replaced locally by the solution $h = 0$, since $h$ cannot become negative. The solution therefore displays a saltwater "nose" at $x = x_n$, where

$$x_n = \max\{x < x_g : p_S(x) + S(x) + \delta b(x) = 0\}. \tag{37}$$

The basic nose solution is given by (34) for $x > x_n$, and by $h = 0$ for $0 < x < x_n$, though such a steady state is not always unique, as we shall discuss shortly.

Examples of the lens and nose cases are shown in Figure 2. (Note that here and in subsequent figures, the aspect ratio is exaggerated and a schematic ice shelf is depicted for illustrative purposes only.) Freshwater is recharged into the aquifer far inland, and discharged closer to the grounding line, with the total recharge and discharge balancing one another in the steady state.

In the nose case, $\partial s/\partial x$ has a discontinuity at $x_n$, leading to a kink in the streamlines and a jump discontinuity in $q_E$ at this point. This is a quirk of the steady state solution, and is only possible because the flux of saltwater is identically zero: as we shall see in Sect. 4.1, the transient solution never reaches the $h = 0$ state from a nonzero initial condition, meaning that this kink in $h$ and the corresponding jump in $q_E$ are smoothed out in practice. However, the presence of this jump is an indication that saltwater intrusion may have non-negligible effects on freshwater exchange with the basal hydrological system by diverting the flow of fresh groundwater.





### 3.2 Periodic forcing

The above steady states can only be realised if the grounding line position $x_g(t)$ and ice overpressure $p_S(x,t)$ remain steady.
However, Antarctic ice streams such as those in the Ross Embayment have undergone significant grounding line movement
on timescales of tens to hundreds of thousands of years. For typical values of the dimensionless hydraulic conductivity $K$ (see
Table 2), the present-day groundwater dynamics are therefore unlikely to have reached a steady state since the most recent
grounding line migration.

We therefore investigate dynamic solutions of Eq. (27), considering in particular periodic solutions. This is motivated by
the fact that global temperatures during the Pleistocene display periodicity of around 100 kyr, associated with variations in
the Earth's orbit of the sun (Willeit et al., 2019), which suggests that a periodic form of the grounding line forcing $x_g(t)$ is
appropriate. In particular, we use a dimensionless forcing

$$x_g(t) = \bar{x}_g - \Delta x_g \cos\left(2\pi t\right), \tag{38}$$

wherein $\bar{x}_g$ and $\Delta x_g$ are constants.

Figure 3, along with Supplementary Animations S1–S3, shows solutions to Eq. (27) for periodic grounding line forcing at
various values of $K$. The solution $z = s(x,t)$ is initialised in a steady state, then calculated over several cycles of advance and
retreat until it reaches a stable periodic solution. Only the final cycle of advance and retreat, which represents this periodic
solution, is plotted.

For all values of $K$, the interface lies between the would-be steady states at the minimum and maximum grounding line
positions within the cycle. Although the forcing is symmetric in time, the interface $z = s(x,t)$ takes different shapes during
advance and retreat.

When $K = 10$ (Figure 3(a)(i)), groundwater flow is fast compared to grounding line movement. The interface therefore
adjusts relatively rapidly, and is consequently close to the quasi-steady state corresponding to the instantaneous grounding line
position. We see that the steady-state solutions are most relevant in the limit of large $K$.

When $K = 1$ (Figure 3(b)(i)), groundwater flow takes place at a moderate speed. There is now a substantial region of
saltwater which lies below the interface $z = s$ throughout the whole cycle, but above the steady-state position of the interface
at the most advanced grounding line position (black dashed line). We can regard this as dynamically "trapped" saltwater, which
would be displaced by freshwater in the steady state for the most advanced grounding line position, but remains present in the
periodic solution due to the unsteadiness of $x_g(t)$. For this particular value of $K$, the lens height $H(0,t) - h(0,t)$ achieves its
maximum near the minimum grounding line position, so that the dynamics of the interface at $x = 0$ are nearly in antiphase with
those at the grounding line. This results in an additional region of trapped *freshwater*, which always lies above the interface
$s(x,t)$, but below its steady state position at the most retreated grounding line position.

Conversely, when $K = 0.1$ (Figure 3(c)(i)), groundwater flow is slow compared to grounding line movement. In this case,
$s(x,t)$ is near-steady throughout most of the domain, apart from a thin freshwater region near the grounding line. The cor-
responding amount of saltwater trapped by the periodic motion is very large, while the corresponding amount of trapped
freshwater is small.



**Figure 3.** Stable periodic solution for $s(x,t)$ during (i) advance and (ii) retreat, and $q_E(x,t)$ during (iii) advance and (iv) retreat, with periodic grounding line forcing given by (38), for (a) $K = 10$ (high permeability) (b) $K = 1$ (moderate permeability) (c) $K = 0.1$ (low permeability). Parameters are as listed in Tables 1 and 2 with $\alpha = 0.1$. For full solutions see Supplementary Animations S1–S3.

Notably, the quasi-steady portion of the solution for small $K$ does not correspond to the steady state at the mean grounding line position $\bar{x}_g$. We hypothesise that as $K \to 0$, this state would approach the steady state for the most retreated grounding





line position. This is because grounding line retreat causes seawater to instantaneously displace any freshwater from the newly

exposed sediments, whereas, during re-advance, the more buoyant freshwater can only displace saltwater in finite time, at a rate controlled by $K$. This is the key cause of asymmetry between grounding line advance and retreat. We infer from this example that the furthest inland grounding line position is most important for determining the extent of saltwater intrusion whenever grounding line movement is fast compared to the natural timescale of groundwater flow.

The exfiltration flux $q_E$ also depends strongly on $K$, as shown in Figures 3(a)–(c)(ii). In addition to scaling roughly linearly

with $K$, $q_E$ is also related to the position of the interface $s$. When $K$ is large, $q_E$ is therefore close to the instantaneous quasi-steady state value, resembling that shown in Figure 2(b)(i). However, when $K$ is small, exfiltration displays a prominent peak near $x_g$ during grounding line retreat, where freshwater is rapidly flushed out of the aquifer due to seawater intrusion. Seawater intrusion therefore significantly affects the basal hydrology via $q_E$, as we shall see again in Sect. 5.

## 4 Solutions in a non-uniform aquifer geometry

### 4.1 "Pocket" steady states

For certain basement geometries $b$ and $S$ and overpressures $p_S$, Eq. (21) also admits steady states which, in addition to a nose, contain saltwater "pockets", where $h > 0$ on a discrete region that is not connected to the saltwater nose. An illustrative example of these "pocket" steady states is shown in Figure 4. These pockets represent another form of saltwater trapping, which can occur in the steady state solution rather than requiring ongoing grounding line movement. For a single pocket in

$x_q < x < x_p$, where $x_p < x_n$, we have

$$p_S(x) + S(x) + \delta s(x) = p_S(x_p) + S(x_p) + \delta s(x_p) \quad \text{for} \quad x_q < x < x_p, \tag{39}$$

$$h = 0, \quad \frac{\partial h}{\partial x} \leq 0 \quad \text{at} \quad x = x_{p-}, \tag{40}$$

where the "$-$" subscript denotes a limiting value taken from the left-hand side, and

$$h = 0 \quad \text{at} \quad x = x_q. \tag{41}$$

Such a pocket is therefore possible if and only if

$$\frac{\partial}{\partial x}(p_S + S + \delta b) \geq 0 \quad \text{at} \quad x = x_p. \tag{42}$$

The pocket solution is then given by

$$h = \frac{1}{\delta}\left[p_S(x_p) - p_S(x) + S(x_p) - S(x)\right] + b(x_p) - b(x) \quad \text{for} \quad x_q \leq x \leq x_p. \tag{43}$$

where $x_q$ is the largest $x < x_p$ for which this expression is zero. We impose that $h = 0$ for $x < x_q$ to avoid the above solution becoming negative, making $x_q$ effectively a second nose (although in general, further pockets could exist in the region $x < x_q$). (It is also possible for a pocket to form a saltwater lens, if the above expression for $h$ is positive for $0 \leq x \leq x_p$ ).



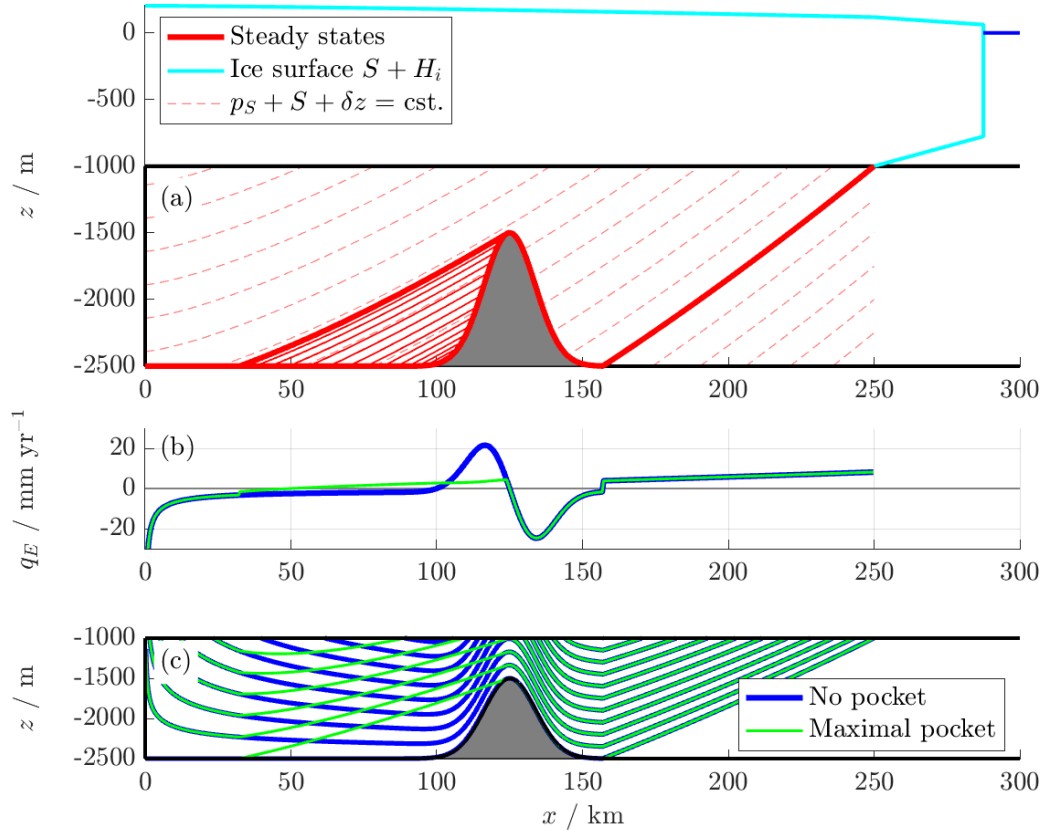

**Figure 4.** (a) A case with infinitely many possible "pocket" steady states, with the criteria (42) illustrated. (b) Freshwater exfiltration and (c) streamlines, with no pocket and the largest possible pocket. Parameters are as listed in Tables 1 and 2, with $k = 10^{-12}$ m$^2$ $K = 0.41$ and $\alpha = 0.05$.

Since $\partial p_S / \partial x < 0$ in a realistic setting, and $|\partial S / \partial x|$ is usually small compared to $|\partial b / \partial x|$ (see Sect. 5), Eq. (42) effectively requires that $\partial b / \partial x$ is positive and relatively large (of order $1/\delta \approx 40$). However, the Dupuit approximation (3) remains valid in such a geometry, since the aspect ratio of the aquifer is around $1/500$ (see Table 2).

In the example of Figure 4, the sedimentary basin contains an idealised bottleneck due to a local rise in the basement rock. Behind this obstacle, the basement $b(x)$ is sufficiently steep for pockets to form via the criterion (42).

We see that the bottleneck in the sediment geometry also produces local exfiltration and subsequent re-infiltration of freshwater, as the flux of freshwater (set by the ice overpressure) is forced through a shallowing and subsequently deepening aquifer. However, the presence of a trapped saltwater pocket extends the distance over which the freshwater region grows shallower, leading to smaller exfiltration over a larger area and substantially modifying the spatial dependence of $q_E$. We infer that potential saltwater pockets, along with saltwater intrusion more generally, may significantly impact the role of sedimentary basins in basal hydrology.



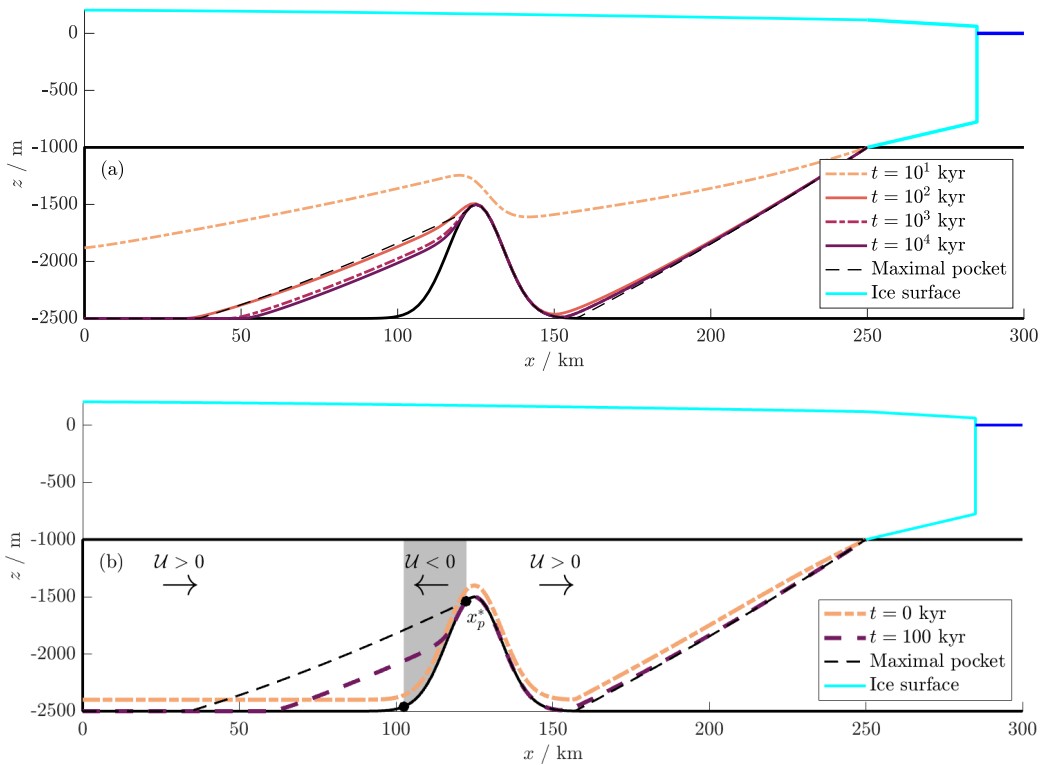

**Figure 5.** (a) Relaxation from an initial state $h = H$ towards a pocket steady state. (b) Relaxation towards a steady state from an initially perturbed pocket-free state, with the direction of $\mathcal{U}(x)$ indicated. Note that $x_p^*$ is not at the maximum of $b(x)$, but a point where $\partial b / \partial x > 0$. Parameters are as in Tables 1 and 2, with $K = 10$ and $\alpha = 0.05$. For full solutions see Supplementary Animations S4 and S5.

When pocket steady states exist, they are non-unique, since we may choose any $x_p$ from the interval on which Eq. (42)
holds, or simply have $h = 0$ for all $x < x_n$. Figure 4 also illustrates this infinite family of steady states and their relation to the criterion (42). Choosing the smallest such $x_p$ leads to the "maximal" pocket, which contains the largest possible volume of trapped saltwater. Depending on the geometry, it may also be possible to satisfy Eq. (42) on multiple disjoint intervals, so that steady states exist with several discrete saltwater pockets. However, it is not yet clear how, if at all, the existence of pocket steady states affects the transient solution of Eq. (27). It is natural to ask, firstly, whether pocket steady states are stable, and, if
so, whether a realistic combination of initial conditions and forcings can lead to such states. We address these questions in the following subsection.

## 4.2 Stability of pocket steady states

For the case of a steady ice sheet $p_S$ and grounding line $x_g$, the transient solution of Eq. (27) relaxes towards a steady state. However, in the case where there are multiple possible steady states, it is unclear which will be achieved. In this and the





following subsection, we demonstrate that a range of final steady states with or without pockets can be obtained, depending on
initial conditions and on the dimensionless hydraulic conductivity $K$.

Figure 5(a) and Supplementary Animation S4 show a transient solution for the geometry and ice sheet shown in Figure
4, in which the aquifer is initially saturated with saltwater throughout ($h = H$). As freshwater enters the aquifer from above,
a significant volume of saltwater is trapped behind the obstacle, resulting in an eventual pocket steady state. However, as
becomes evident after a very long period of time, the final state does not have the largest possible pocket, as might be expected.
This overshoot occurs because the slope of the steady interface $\partial s/\partial x$ fails to be continuous at $x_p$ and $x_q$, where the solution
switches between $h = 0$ and $h > 0$. On the other hand, the transient solution $s(x,t)$ has a continuous derivative due to the
regularity imposed by Eq. (27). As a result, this solution is smoothed compared to the steady state, as seen for instance at $10^2$
kyr. The smoothed region near $x_p$ in turn allows a small amount of trapped saltwater to escape the pocket.

We next consider the opposite case of a small initial perturbation to the pocket-free state, as shown in Figure 5(b) and
Supplementary Animation S5. Near $h = 0$, the linearised version of Eq. (21) takes the form

$$\frac{\partial h}{\partial t} + \frac{\partial}{\partial x} \left[ \mathcal{U}(x)h \right] = 0, \tag{44}$$

where

$$\mathcal{U}(x) = -K \frac{\partial}{\partial x} \left( p_S + S + \delta b \right). \tag{45}$$

This is a hyperbolic equation representing advection with velocity $\mathcal{U}$. Equation (45) indicates that, at leading order, the flow is
driven by the overburden and topographic pressure gradients, and the second-order buoyancy term is negligible.

The direction in which a perturbation (i.e. a small volume of saltwater) travels is determined by the sign of $\mathcal{U}$, which is shown
in Figure 5(b). The condition (42) for a possible pocket in $x_q < x < x_p$ is equivalent to $\mathcal{U}(x_p) < 0$, and the largest possible
pocket occupies $x_q < x < x_p^*$, where $\mathcal{U}(x_p^*) = 0$. Therefore, a small volume of saltwater in $x < x_p^*$ close to $x_p^*$ will move in the
negative $x$-direction and become trapped in the region $x < x_p^*$. This trapped saltwater will in turn eventually form a pocket in
the final steady state. Meanwhile, a small volume of saltwater in $x_p^* < x < x_n$ will travel in the positive $x$-direction and reach
the saltwater nose.

To illustrate this, Figure 5(b) and Supplementary Animation S5 show the result of initially perturbing a pocket-free state.
The eventual steady state towards which the solution relaxes includes a pocket of trapped saltwater. The volume of saltwater
in $0 < x < x_p^*$, given by the integral of $h(x,t)$ over this region, changes by less than 2% from the initial to final state, strongly
suggesting that this pocket has been formed by trapping the saltwater initially located in this region, via the linearised flow $\mathcal{U}$.
Conversely, the saltwater in $x > x_p^*$ has been evacuated towards the ocean.

### 4.3   Pocket creation by grounding line advance

If grounding line movement brings about a change from a system with a unique steady state to one with multiple steady states,
it is unclear towards which of the new available steady states the solution of Eq. (27) will relax. Here we show that it is possible
for the system to reach either a pocket state or one without, depending on the intermediate states, and on the parameter $K$.



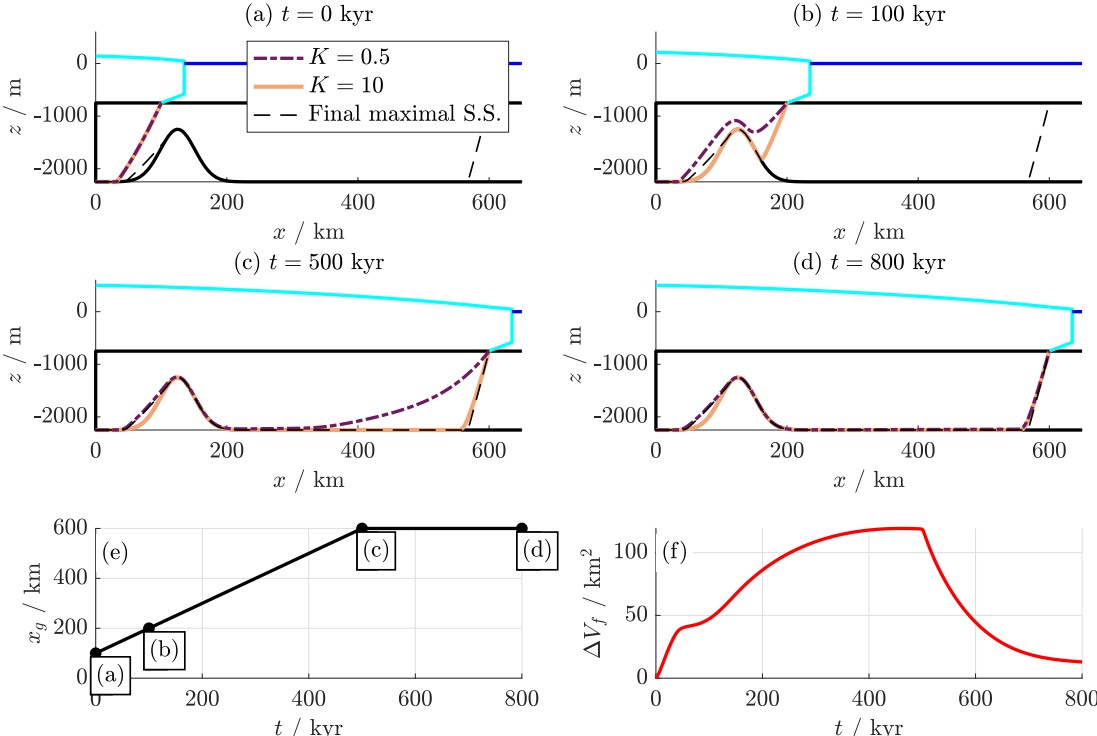

**Figure 6.** (a–d) Solutions $z = s(x, t)$ plotted at various points in time for two different $K$. (e) Grounding line position $x_g$ (f) Difference in freshwater volumes $\Delta V_f(t) = V_{f, K=10} - V_{f, K=0.5}$ between the two cases. Parameters are as listed in Tables 1 and 2, with $\alpha = 0.1$. For full solutions see Supplementary Animation S6.

For the purposes of plotting our results, we also define the total volume of freshwater:

$$V_f = \int_0^{x_g} H - h \, dx. \tag{46}$$

Figure 6 and Supplementary Animation S6 include an example wherein the grounding line advances from an inland position past a bottleneck similar to that shown in Figure 4. The final grounding line position accommodates multiple steady states, although only the maximal and minimal such states are shown. Different final steady states are reached depending on the relative speed of groundwater flow, described by the parameter $K$.

When groundwater flow is sufficiently slow compared to grounding line movement (e.g. for $K = 0.5$), the interface slowly relaxes from above to the maximal pocket state, resulting in trapped saltwater. On the other hand, when groundwater flow is 355 fast (e.g. for $K = 10$), the interface rapidly relaxes to the quasi-steady state for the instantaneous grounding line position. If one of these intermediate quasi-steady states inundates the region of the would-be pocket with freshwater, the final steady state has no pocket and no saltwater is trapped.





**Figure 7.** Measurements of (a) upper surface $S$ (b) basement $b$ (c) thickness $H$ for the Ross Sea sedimentary basin, which we sample along transects (Fretwell et al., 2013; Rignot et al., 2013; Greene et al., 2017; Mouginot et al., 2017; Kingslake et al., 2018; Gustafson et al., 2022; Tankersley et al., 2022). (d) Averaged cross-section along transects.

Similarly to the case of periodic forcing in Sect. 3.2, we see from this example that the trapping of saltwater (in this case via a pocket steady state) is dependent on the dimensionless hydraulic conductivity $K$ as well as previous grounding line
forcing. Indeed, this model demonstrates the possibility of "hysteresis", where grounding line retreat and re-advance induces a transition between two possible steady states.



## 5 Application: Ross Sea, West Antarctica

### 5.1 Design

Having explored the above model of seawater intrusion in idealised settings, we now seek to apply it to a real-world scenario,
using the sedimentary basin beneath the Ross Ice Shelf in West Antarctica as a case study. As well as being the location
of observed paleo-seawater intrusions by Gustafson et al. (2022), the thickness of this sedimentary basin has been mapped
in detail by radar measurements by Tankersley et al. (2022), and the grounding line history of this region has been subject
to extensive investigation and modelling (Venturelli et al., 2023, 2020; Neuhaus et al., 2021; Lowry et al., 2019; Kingslake
et al., 2018). Since the data are three-dimensional, we investigate a transect of the data corresponding roughly to a flowline
of Whillans ice stream (more precisely, we average between a set of adjacent transects to reduce uncertainty due to lateral
variation). A summary of the basement geometry and grounding line data used, along with our choices of transect, and the sites
of magnetotelluric measurements of salinity by Gustafson et al. (2022), is plotted in Figure 7.

### 5.2 Aquifer geometry

Our estimates for $H(x)$ and $b(x)$ along the selected flowline come from radar measurements by Tankersley et al. (2022), which
we combine with satellite data for $S(x)$ from Fretwell et al. (2013). Since the measurements of $b$ and $H$ only extend a short
distance beyond the present-day grounding line, we linearly extrapolate $b$ in this region, assuming that $H(0) = 0$ based on
localised seismic measurements (Peters et al., 2006). The onset of the sedimentary basin $x = 0$ corresponds to the onset of ice
streaming, which can be inferred from surface ice flow velocity. We also select a point on the flowline, marked with a red cross,
corresponding to the Subglacial Lake Whillans (SLW) observation site, where the salinity of the sedimentary basin has been
measured by Gustafson et al. (2022).

### 5.3 Grounding line and ice sheet model

Our reconstruction of the grounding line position $x_g(t)$ is based on paleo-ice-sheet modelling supported by geophysical obser-
vations, which suggests (Kingslake et al., 2018):

1. Slow advance to a maximum, located at the edge of the Antarctic continental shelf, during 100–20 kyr before present,

2. Rapid retreat to a minimum, several hundred kilometers inland of the present-day grounding line position, during 20–10
   kyr before present,

3. Re-advance from this minimum to the present-day grounding line during 10–0 kyr before present.

Since there is substantial uncertainty in the precise speed and timing of grounding line advance and retreat, we simply linearly
interpolate $x_g(t)$ between the maximum, minimum and present-day position, as shown in Figure 10(a) and 11(a).



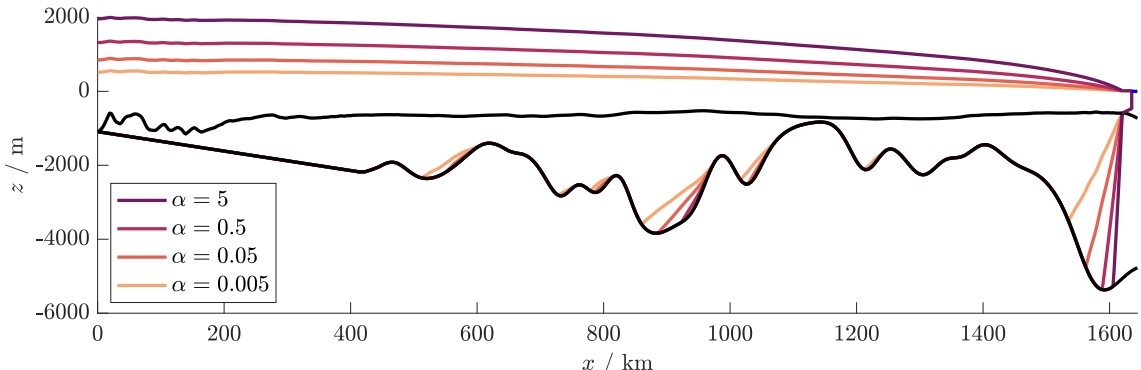

**Figure 8.** Potential steady state pockets for the basement geometry in Figure 7, for various values of the ice sheet parameter $\alpha$. (Any pockets in the region ($x < 400$ km) where $b(x)$ has been linearly interpolated are ignored.) Parameters are as listed in Tables 1 and 2.

We assume that the grounding line behaviour obeys 100-kyr periodicity, in accordance with the periodic behaviour of global temperatures during the Pleistocene (Pollard and DeConto, 2009; McKay et al., 2012). We therefore repeat our approach from Sect. 3.2 wherein the model is run for a series of forcing cycles until a stable periodic solution is reached.

To model $p_S$, we employ the steady shallow ice sheet model described in (28), with the constant $\alpha = 5$ based on the results of paleo-ice sheet reconstructions using more sophisticated models (Huybrechts, 2002; Kingslake et al., 2018). We choose to use this simplified model on the basis that the exact form of $p_S$ is a relatively small source of model uncertainty compared to e.g. lateral variations in $H$ and $b$, or heterogeneity in aquifer properties such as $k$ and $\phi$.

### 5.4 Aquifer properties and timescales

The largest source of uncertainty is given by the hydraulic properties of the aquifer, namely $k$ and $\phi$. This includes model uncertainty (e.g. possible spatial heterogeneity in $\phi$ and $k$, and dependence of $k$ on $\phi$) and parameter uncertainty (assuming that $k$ is constant, what should its value be?). Previous studies have accounted for this by solving models for a wide range of possible $k$, typically $10^{-17}$–$10^{-13}$ m$^2$ (Gooch et al., 2016; Li et al., 2022; Robel et al., 2023). In the dimensionless model, the values of $k$ and $\phi$ are both accounted for in the single parameter $K$, which takes small to moderate values for the above range of $k$ (see Table 3). Note that we disregard values larger than $k = 3 \times 10^{-14}$ m$^2$, which gives $K \approx 0.01$, on the basis we do not expect solutions for very small values of $K$ to differ substantially, and that we include an artificially high value $k = 1 \times 10^{-10}$ m$^2$ ($K \approx 40$) for comparison. In Appendix B, we discuss how Eq. (21) can be generalised to the case of variable $k(x,z,t)$ and $\phi(x,z,t)$.

### 5.5 Results

Figure 8 shows where potential steady-state saltwater pockets can exist in the above geometry for a range of ice thicknesses, which in our model are characterised by the parameter $\alpha$. For our choice of $\alpha = 5$, which is based on historic ice thickness



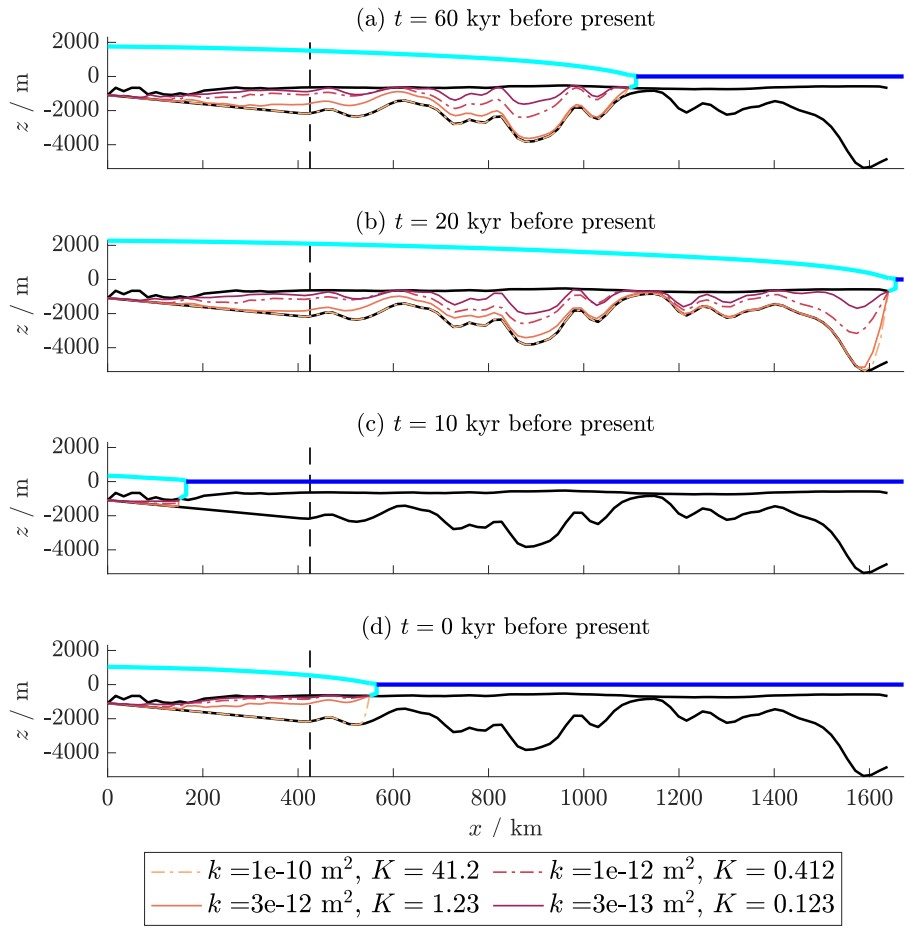

**Figure 9.** Interface $s(x,t)$ for varying $k$, plotted (a) during grounding line advance (b) at grounding line maximum (c) at grounding line minimum (d) at present day, with the SLW site marked by a black dashed line. Parameters are as listed in Tables 1 and 2, with $\alpha = 5$. For full solutions see Supplementary Animations S7–S12.

**Table 3.** Depth of the freshwater lens at the SLW site for various values of $k$.

| $k$ / m$^2$ | $K$ | $d_{f,\text{SLW}}$ / m |
|---|---|---|
| $1 \times 10^{-10}$ | 41 | 1520 |
| $3 \times 10^{-12}$ | 1.2 | 480 |
| $1 \times 10^{-12}$ | 0.41 | 200 |
| $3 \times 10^{-13}$ | 0.12 | 150 |
| $1 \times 10^{-13}$ | 0.041 | 140 |
| $3 \times 10^{-14}$ | 0.012 | 130 |



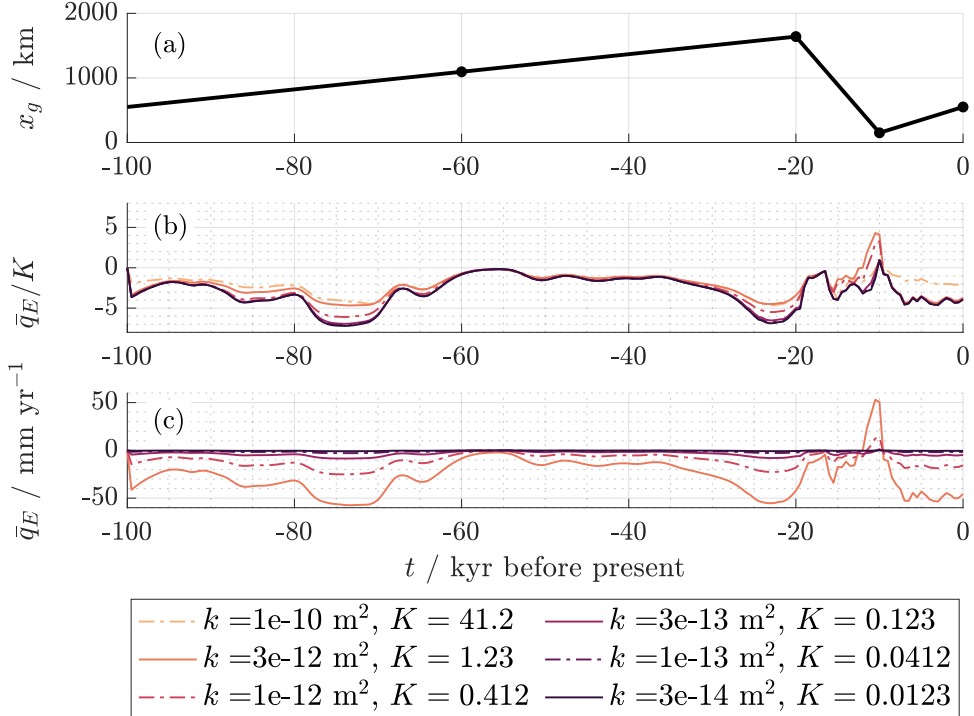

**Figure 10.** (a) Grounding line position $x_g(t)$. (b) Dimensionless spatially averaged exfiltration $\bar{q}_E$ after scaling with $K$. (c) Dimensional $\bar{q}_E$. Parameters are as listed in Tables 1 and 2, with $\alpha = 5$.

modelled by Kingslake et al. (2018) and Huybrechts (2002), the ice surface is too steep to permit saltwater pockets from existing in the steady state. However, such pockets are possible for shallower ice sheets, or in the transient case.

The results of our time-dependent simulations are shown in Figures 9, 10 and 11 and Table 3, as well as Supplementary Animations S7–S12. For each value of $k$, we calculate the present depth of the freshwater lens at the SLW site

$$d_{f,\text{SLW}} = S(x_{\text{SLW}}) - s(x_{\text{SLW}}, 0),\tag{47}$$

along with the space-averaged exfiltration

$$\bar{q}_E(t) = \frac{1}{x_g} \int\limits_0^{x_g(t)} q_E(x,t)\,\mathrm{d}x.\tag{48}$$

Figure 9 shows the interface $s(x,t)$ at selected points in time for various values of $k$, and Table 3 shows the corresponding values of $d_{f,\text{SLW}}$. Figure 10 shows a time series of the space-averaged exfiltration $\bar{q}_E$, both dimensionally and dimensionlessly after scaling with $K$, and Figure 11 shows a space-time plot of $q_E(x,t)$ for the case $k = 3 \times 10^{-12}$ m$^2$ ($K = 1.2$), alongside the basement geometry for comparison.



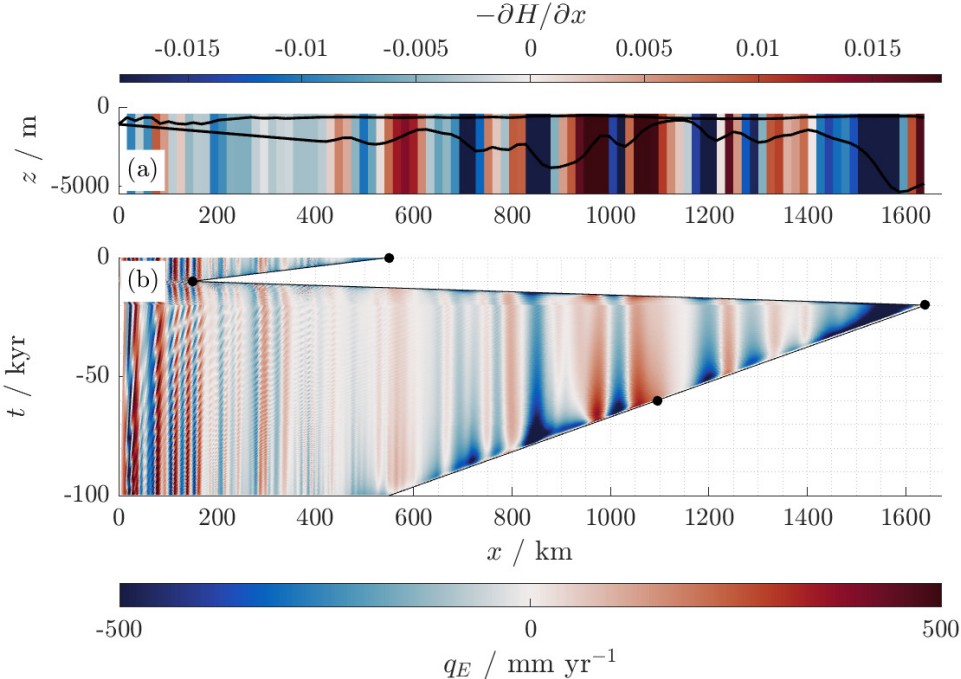

**Figure 11.** (a) Basement geometry $b(x)$ and $S(x)$, coloured according to $-\partial H/\partial x$. (b) Space-time plot of $q_E(x,t)$, with red corresponding to $q_E(x,t) > 0$ (exfiltration) and blue to $q_E < 0$ (infiltration). Values with $|q_E| > 500$ mm yr$^{-1}$ are plotted using the same colours as $\pm 500$. Parameters are as listed in Tables 1 and 2, with $\alpha = 5$ and $k = 3 \times 10^{-12}$ m$^2$.

For the very highly permeable test case ($k = 10^{-10}$ m$^2$) in Figure 9, the interface is close to the quasi-steady state for the current grounding line position, as expected. The resemblance to the steady state is also observed for $k = 3 \times 10^{-12}$ m$^2$, in addition to a thin layer of trapped saltwater. For the successively lower permeabilities, the depth of the transient freshwater lens becomes successively less, and correspondingly more saltwater is trapped due to grounding line movement. For the less permeable examples, we see that that both possible solution forms within the complementarity problem (27) are required, as the solution exhibits small but significant regions of saltwater exfiltration where $h = H$ (for example, near $x = 600$ m and $x = 1200$ m in Figure 9(b)).

Table 3 shows the present-day depth of the transient freshwater lens at the location of SLW in the above solutions. This increases with the permeability $k$, reaching the full depth $H$ of the layer for the very high value of $10^{-10}$ m$^2$ where the solution is near quasi-steady. Field measurements by Gustafson et al. (2022) indicate that groundwater becomes fully saline around 600m below the ice bed (albeit with a substantial region where freshwater and saltwater mix), which is most closely matched by the solution for the high permeability $k = 3 \times 10^{-12}$ m$^2$.





The exfiltration $q_E$ scales roughly linearly with $K$ (see Eq. (26)), but also depends on $K$ through $s(x,t)$. Figure 10(b) allows us to see the importance of the latter dependence: if $q_E$ were determined by basement geometry and ice overpressure alone (i.e.

if seawater intrusion were ignored), the curves would collapse onto one another. However, the magnitude of $\bar{q}_E/K$ is at points several times larger for $k = 3 \times 10^{-14}$ m$^2$, which includes the most trapped saltwater, than for $k = 10 \times 10^{-10}$ m$^2$, where effectively no saltwater is trapped. We therefore note that saltwater intrusion and trapping may have a substantial effect on groundwater exchange with the shallow hydrological system. We also note that $\bar{q}_E < 0$ throughout most of the cycle, meaning that, on average across the basin, more water infiltrates than is exfiltrated (ultimately driven by the downstream hydraulic

potential gradient due to the slope of the ice surface). However, around the glacial minimum exfiltration dominates. Over the duration of a periodic cycle, the net flux of freshwater into the domain must be zero, which is achieved by localised discharge of freshwater at the grounding line $x_g$ during retreat.

Figure 10(c) allows us to compare the dimensional values of $\bar{q}_E$. For the large values $k = 3 \times 10^{-12}$ m$^2$ and $k = 1 \times 10^{-12}$ m$^2$, the magnitude of $\bar{q}_E$ is in the low tens of millimeters per year, dropping to a few millimeters per year for lower permeabilities.

However, local values of $q_E$ may be several times larger than the average $\bar{q}_E$, as seen in Fig. 11. Previous modelling has found that groundwater contributes significantly to subglacial groundwater budgets (Christoffersen et al., 2014). This suggests that a realistic value of $q_E$ should be similar in size to other sources of basal water, such as the basal melt rate, which may reach 20–100 mm yr$^{-1}$ beneath an ice stream (Joughin et al., 2009). To achieve a mean exfiltration of this order of magnitude, we require high permeabilities $k \gtrsim 10^{-12}$ m$^2$, which is consistent with our previous estimate based on freshwater lens depth. For

smaller permeabilities, such values of $q_E$ are locally achievable, but are counterbalanced by infiltration ($q_E < 0$) elsewhere.

Moreover, any estimate obtained under the assumptions of our model will fail to consider short-term exfiltration due to poro-elastic deformation of the sediments, which previous modelling suggests may reach tens to hundreds of mm yr$^{-1}$ for $k = 1 \times 10^{-13}$ m$^2$ (Li et al., 2022; Robel et al., 2023). This short-term exfiltration may enable sedimentary basins to act as a net source of basal water, as is suggested by the modelling of Christoffersen et al. (2014), rather than a sink. However, further

modelling is required to understand the combined effects of short-term and long-term exfiltration.

In addition, $q_E$ is highly heterogeneous in space, as seen in Figure 11, displaying localised regions of significant exfiltration and infiltration. The sign of $q_E$ is closely correlated to that of $-\partial H/\partial x$, with exfiltration generally occurring where the aquifer is shallowing in the direction of flow and infiltration where it is deepening. This correlation occurs because the terms $\partial S/\partial x$ and $\delta\partial s/\partial x$ are relatively small in (26), so that we have

$$q_E \approx K\frac{\partial}{\partial x}\left(H\frac{\partial p_S}{\partial x}\right) \approx -K\frac{\partial H}{\partial x}\left|\frac{\partial p_S}{\partial x}\right|, \tag{49}$$

with the latter approximation holding because $\partial p_S/\partial x$ varies relatively little compared to $H$ throughout the aquifer. The striped pattern of Figure 11 indicates that this term dominates throughout most of the domain over time. However, there are also visible deviations from this pattern due to the other terms, such as the role of saltwater intrusion as discussed above. These are most prominent near the grounding line, where the interface $s(x,t)$ and ice thickness $H_i(x,t)$ are steepest.

Overall, we therefore find that a high permeability $k \gtrsim 10^{-12}$ m$^2$ is required to reproduce both the observed depth of the freshwater lens at the SLW site and is consistent with the modelled contribution of groundwater to the basal water budget.



This is an unusually high value for a sedimentary rock, suggesting that geological features such as faults and fractures, which our model has not considered explicitly, but which may macroscopically be captured by a large permeability, may play an important role in subglacial groundwater transport. Our chosen value of $k$ represents an idealisation of a sedimentary aquifer
which may in reality be highly heterogeneous.

## 6    Conclusions

In this paper, we have developed a mathematical model for groundwater flow and seawater intrusion in the sedimentary basin beneath a marine ice sheet, considering in particular the effects of grounding line advance and retreat and of sedimentary basin geometry. Both of these lead to the phenomenon of saltwater trapping, where saltwater is permanently resident in regions
where a simplistic steady-state solution would predict freshwater. Saltwater trapping is also highly dependent on hydraulic properties, which we have accounted for with the parameter $K$. The concept of saltwater trapping, via pocket steady states, should generalise from a subglacial context to a wide range of seawater intrusion problems provided that the aquifer geometry is sufficiently nonuniform. We discuss this generalisation in Appendix B.

In order to understand the role of groundwater in the subglacial hydrological system, it is crucial to constrain the dimension-
less hydraulic conductivity $K$, since the flux $q_E$ of water from the aquifer to the shallow hydrological system scales with this value. However, the appropriate value of $K$ is highly uncertain, due to the observational challenges in measuring the porosity and permeability of such an aquifer, along with the fundamental limitations of modelling such an aquifer as homogeneous and isotropic (Appendix B discusses how this assumption may be relaxed). However, by modelling groundwater flow beneath the Ross Ice Shelf, using our knowledge of grounding line history, we are able to gain insight into the effect of $K$ on saltwater
trapping. By combining this modelling with present-day observations of trapped saltwater, we are able to estimate a physically appropriate range of values for $K$. We find that high values of the permeability ($k \sim 10^{-12}$ m$^2$) are required to reproduce observations of the freshwater lens depth, and are also consistent with modelled contributions of groundwater to the basal water budget via exfiltration.

Our modelling also suggests that the spatial exfiltration $q_E$ of groundwater into the shallow hydrological system is dominated
by sedimentary basin geometry. Groundwater is generally exfiltrated where the aquifer is thinning in the direction of flow, and infiltrates where the aquifer is thickening. However, the ice overpressure $p_S(x,t)$ and the freshwater-saltwater interface $s(x,t)$ also affect $q_E(x,t)$. In the latter case, the presence of trapped or intruding saltwater changes the shape of the freshwater-saturated region of the aquifer, and in some cases may entirely displace freshwater so that saltwater is exfiltrated into the groundwater system.

The latter possibility highlights a limitation of our model, namely our idealised treatment of the shallow hydrological system in assuming that only freshwater re-infiltrates the aquifer and that the effective pressure $p_e = 0$. A more complex model, relating $q_E$ and $p_e$ by accounting for the dynamics of the shallow hydrological layer, and considering both freshwater and saltwater, would provide a more subtle understanding of the role of groundwater exfiltration and infiltration in subglacial hydrology.





Moreover, another possible development of this model could consider the effect of exfiltration and infiltration on subglacial
heat transport. Previous modelling by Gooch et al. (2016) suggests that exchange of groundwater with the subglacial hydro-
logical system may substantially affect geothermal heat flux to the ice bed, and subsequently basal melting. Tankersley et al.
(2022) note that sedimentary basin geometry, in particular the presence of highly permeable faults, may affect ice sheet dynam-
ics via the effect of groundwater on heat flux and on basal melting. Such a model would require an equation of heat transport
via advection and diffusion, and could include freshwater generation via basal melting of the ice. For a groundwater flux scale
of $[q_E] = 10$ mm yr$^{-1}$, a porous medium conductivity of $k_c = 3$ W m$^{-1}$ K$^{-1}$ and a specific heat capacity of water $c_w = 4000$
J kg$^{-1}$ K$^{-1}$ (Gooch et al., 2016), the Péclet number associated with vertical heat transport is

$$\text{Pe} = \frac{[q_e][z]\rho_f c_w}{k_c} \approx 0.4, \tag{50}$$

which is $O(1)$, suggesting that both advection by groundwater and diffusion are important to subglacial vertical heat transport.

In addition, we have assumed that the aquifer is rigid, and neglected the dynamic response of the sediments to loading.
This enables us to model background rates of exfiltration and infiltration over the long timescales associated with grounding
line movement and horizontal groundwater flow. However, a full account of the exchange of groundwater with the shallow
hydrological system at a given point in time should also include exfiltration due to dynamic effects, namely the poro-elastic
response of sedimentary basins to ice sheet loading, which has been separately modelled by Gooch et al. (2016), Li et al.
(2022), and Robel et al. (2023). Indeed, short-term exfiltration may substantially exceed the long-term value: for example, Li
et al. (2022) and Robel et al. (2023) obtain short-term exfiltration rates of 10–100 mm yr$^{-1}$ for $k = 10^{-13}$ m$^2$, compared to
our modelled average $\bar{q}_E \approx -2$ mm yr$^{-1}$ for the same $k$.

Our model has made the assumption of a sharp interface dividing freshwater and saltwater. This greatly simplifies the task
of solving the problem, at the cost of being unable to account for the smoothly varying salinity observed in the measurements
of Gustafson et al. (2022). The sharp-interface assumption is warranted provided that certain Péclet numbers corresponding
to mixing by molecular diffusion and hydrodynamic dispersion are large (Dentz et al., 2006; Bear, 2013; Koussis and Mazi,
2018). To account for the effect of mixing, it is possible (though more complicated) to solve the full problem of density-coupled
salt transport, known as the Henry problem, (e.g. Croucher and O'Sullivan (1995)) or to introduce a "mixing layer" around the
sharp interface where these effects are accounted for (Van Duijn and Peletier, 1992; Paster and Dagan, 2007). The process of
saltwater mixing has the potential to mitigate saltwater trapping, for example by entraining saltwater from a pocket into the
freshwater flow.

In conclusion, we have used an idealised model to demonstrate several previously unidentified features of a classical sea-
water intrusion problem which emerge when attempting to model groundwater flow beneath a marine ice sheet, namely the
trapping of saltwater via grounding line movement and aquifer geometry. Our work provides new insights into the modelling of
groundwater flow in subglacial sedimentary basins on geological timescales, and the contribution of these sedimentary basins
to Antarctic subglacial hydrology. Using this model, we are able to make quantitative estimates of groundwater exfiltration and
infiltration throughout space and time, and to infer hydraulic properties of the aquifer based on local measurements of salin-
ity. Our work provides a starting point for more developed models of groundwater flow in sedimentary basins, which could




include a more sophisticated treatment of hydraulic properties, three-dimensional flow, coupling groundwater flow to shallow hydrology and the ice sheet itself, or quantification of the uncertainty in modelling the above. Further modelling is required to advance our understanding of sedimentary basin hydrology, and its role in determining the stability of the West Antarctic Ice Sheet.

**Appendix A: Numerical method**

In order to numerically solve Eq. (21) on the changing domain $0 < x < x_g(t)$, we apply the coordinate transformation

$$x = x_g(t)X. \tag{A1}$$

The rescaled equation (after rearrangement into a more canonical conservation form becomes

$$\frac{\partial h}{\partial t} = \frac{\partial}{\partial X}\left[K\frac{h}{x_g^2}\frac{\partial}{\partial X}(p_S + S + \delta s) + \frac{\dot{x}_g}{x_g}Xh\right] - \frac{\dot{x}_g}{x_g}h, \tag{A2}$$

which may then be solved on the fixed domain $0 < X < 1$ subject to the conditions (17) and (18). To solve (A2) as part of the more general complementarity problem (27), we formulate the discrete complementarity problem

$$U^2 = -\frac{\partial h}{\partial t} + \frac{\partial}{\partial X}\left[K\frac{h}{x_g^2}\frac{\partial}{\partial X}(p_S + S + \delta s) + \frac{\dot{x}_g}{x_g}Xh\right] - \frac{\dot{x}_g}{x_g}h, \tag{A3}$$

$$V^2 = H - h, \tag{A4}$$

$$UV = 0. \tag{A5}$$

This formulation replaces (21) with the more general "complementarity condition" (A5), and Equations (A3) and (A4) enforce the constraints that their respective right hand sides are non-negative. This formulation ensures that either Eq. (A2) is obeyed and $h \leq H$, or $h = H$ and the value $\partial h/\partial t$ implied by (A2) is non-negative (equivalent to $q_E \geq 0$ by Eq. (25)). If this implied value of $\partial h/\partial t$ is negative (equivalent to $q_E < 0$), the solution must detach from $h = H$.

We solve the system (A3), (A4), (A5) using a finite volume method, which retains the conservative properties of Eq. (A2). The "diffusive" buoyancy term multiplied by $\delta$ is discretised using a second-order midpoint arithmetic mean, whereas the remaining "advective" flux terms are discretised using a first-order upwind method. The calculated flux terms may then be used to find $q_E$ via Eq. (25). The time derivative is discretised using a first-order fully implicit scheme. The discrete equation at each timestep is solved using MATLAB's inbuilt function `fsolve`. Between 100 and 200 spatial meshpoints are used, with timesteps between $\Delta t = 0.001$ and $0.01$, both depending on the specific problem in question. (Larger timesteps up to $\Delta t = 1$ are used for the late stages of the long-term simulation shown in Figure 5(a).) Smaller timesteps are required during periods of faster flow, e.g. rapid grounding line movement, to ensure stability of the first-order upwind method.





## Appendix B: Developments of the model

We have presented the above model with a number of simplifying assumptions, in order to illustrate the key features introduced by aquifer geometry and grounding line movement. However, several generalisations of this model are possible, a few of which we outline below.

**B1  Prescribed recharge / unconfined aquifer:**

Our derivation of the above model assumes that $p_S$, the pressure at $z = S$ is known and given by the ice sheet overburden, which is consistent with the existence of a shallow layer of basal water in which the effective pressure is zero. However, it is also possible to write a model in which the exfiltration (or infiltration) $q_E$ is prescribed, and $p_S$ must be found. In this case, Eq. (25) becomes an equation for $p_S$:

$$H\frac{\partial}{\partial x}(p_S + S) + h\delta\frac{\partial s}{\partial x} = \int_0^x q_E\, \mathrm{d}x =: -Q, \tag{B1}$$

using the no-flux condition at $x = 0$. This can then be substituted into Eq. (21) to obtain

$$\frac{\partial h}{\partial t} = K\frac{\partial}{\partial x}\left[h\left(-\frac{Q}{H} + \delta\left(1 - \frac{h}{H}\right)\frac{\partial s}{\partial x}\right)\right]. \tag{B2}$$

This equation displays the same features as Eq. (21). The ice overburden becomes entirely decoupled in this limit, only appearing implicitly through the basal hydrological system, which exchanges freshwater with the aquifer in $0 < x < x_g$. Between the

cases of prescribing $p_S$ or $q_E$ on $z = S$, there are also potential intermediate cases, where a relationship between $p_S$ and $q_E$ is prescribed in order to encapsulate the dynamics of the shallow basal hydrological system.

When considering a glacier that is not marine-terminating (e.g a continental ice sheet during a glacial period), we would also encounter the possibility of an unconfined and unsaturated region of the aquifer between the ice sheet margin and the ocean. This region could also include permafrost, which substantially reduces aquifer permeability.

The region beneath the ice sheet is modelled as in this paper, with a confined aquifer in $b < z < S$, and the pressure $p = p_S$ prescribed on the upper surface. To model the unsaturated region, we introduce a free boundary $z = \hat{S}(x,t) = b + \hat{H}$, representing the water table which lies below the ground surface $z = S$. On $z = \hat{S}$, both the exfiltration $q_E < 0$ (representing rainfall or snow melt) and the pressure $p_S = p_{\mathrm{atm}}$ are prescribed. The resulting model consists of the two equations

$$\frac{\partial h}{\partial t} = K\frac{\partial}{\partial x}\left[h\frac{\partial}{\partial x}\left(\hat{S} + \delta s\right)\right], \tag{B3}$$


$$\frac{\partial \hat{H}}{\partial t} = K\frac{\partial}{\partial x}\left[\hat{H}\frac{\partial \hat{S}}{\partial x} + h\delta\frac{\partial s}{\partial x}\right] - q_E, \tag{B4}$$

with the fluxes of freshwater and saltwater both zero at $x = 0$, and $h = \hat{H} = H$ at the coastline $x = x_c$. Moreover, we must have $\hat{H} = H$ at the ice margin $x = x_m$ (where $H_i(x_m(t),t) = 0$), so that the water table is continuous with the subglacial region.



This problem is more similar to classical seawater intrusion problems, yet still permits multiple steady states for variable $b$. However, since $x_c$ changes only through sea level changes and not through the ice sheet response to climatic forcing, the variation in $x_c$ is less pronounced than that of $x_g$.

## B2   Aquifer heterogeneity

When the porosity $\phi(x,z)$ and permeability $k(x,z)$ are heterogeneous, it is straightforward to derive the generalised form of Eq. (21):

$$\frac{\partial}{\partial t}\left(h\bar{\phi}_h\right) = K\frac{\partial}{\partial x}\left[h\bar{k}_h\frac{\partial}{\partial x}\left(p_S + S + \delta s\right)\right], \tag{B5}$$

where

$$\bar{\phi}_h = \frac{1}{h}\int_b^s \phi\,\mathrm{d}z, \quad \bar{k}_h = \frac{1}{h}\int_b^s k\,\mathrm{d}z. \tag{B6}$$

For example, $\phi$ may display an "Athy profile", in which (Gustafson et al., 2022; Gooch et al., 2016)

$$\phi = \phi_0 e^{-\beta(S-z)}, \quad \bar{\phi}_h = \phi_0 e^{-\beta H}\frac{e^{\beta h}-1}{\beta h}. \tag{B7}$$

The permeability $k$ could be piecewise constant, representing different sedimentary strata (Li et al., 2022), or obey a relation $k(\phi)$ (Freeze and Cherry, 1979). The porosity $\phi$ may also be time-dependent, for example due to sediment compaction.

## B3   Three-dimensional model

In reality, coastal aquifers are three-dimensional structures. The three-dimensional analogue of Eq. (21) is

$$\frac{\partial h}{\partial t} = K\nabla\cdot\left[h\nabla\left(p_S + S + \delta s\right)\right], \tag{B8}$$

where $\nabla = (\partial_x, \partial_y)$ is the horizontal gradient. This equation should be solved on a region $R$, whose boundary comprises an inland ice divide $\partial R_i$, on which $\nabla h\cdot n = 0$, and a grounding line $\partial R_g$, on which $h = H$. As before, the steady states are given by

$$h = 0 \quad \text{or} \quad p_S + S + \delta s = \text{const.} \tag{B9}$$

However, it is not as straightforward to establish a criterion for saltwater pockets analogous to Eq. (42), as this requires knowledge of the global topology of steady state solutions. We must first find the "minimal" steady state, consider the subset of $R_0 \subset R$ on which this state has $h = 0$, and determine whether there exists a constant $C$ such that $p_S + S - C > \delta b$ on a nonempty compact subset of $R_0$.

*Code and data availability.* Supplementary animations for this article are available at https://doi.org/10.5281/zenodo.13759494 (Cairns, 2024b). Codes used to produce the figures and data in this article, and to produce the supplementary animations, are available at https://doi.org/10.5281/zenodo.13759411 (Cairns, 2024a).



*Author contributions.* GJC, GPB and IJH developed the model. GJC wrote the code and produced the results. GJC wrote the paper with input from GPB and IJH.

*Competing interests.* The contact author has declared that none of the authors has any competing interests.

*Acknowledgements.* GC acknowledges the support of an ESPRC doctoral studentship.



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
