# Peer review of "Groundwater dynamics beneath a marine ice sheet"

_EGUsphere, 2024_

## Author Response (AR1)

Dear Dr McCormack,

Thank you for inviting us to submit a revised version of our manuscript. Please find below a point-by-point summary of the revisions we have made in response to reviewer recommendations, and other comments. Page and line numbers refer to the marked-up version of the manuscript.

**Reviewer 1:**

*"The strength of the work is a simple and elegant mathematical design. There are, however some limitations, notably the use of the shallow ice approximation, which is a poor choice in the Siple Coast test case because the fast motion of glaciers there is almost exclusively caused by basal slip. The authors offer a discussion of other model limitations, but not this one. I doubt the model reproduces the actual geometry of the Siple Coast, but this is perhaps not so important, given the "first-order" nature of the study more generally."*

This is true. The focus of this study was intended to be on the groundwater dynamics rather than the mechanics of the ice itself (we are currently working on a coupled model that includes a more sophisticated model for the latter). We required an ice sheet geometry for different grounding-line positions, and we selected the shallow ice model for the sake of simplicity. For the periodically advancing and retreating ice sheet considered in our study, where the groundwater dynamics are far from a steady state, the precise shape of the ice sheet has little effect on the freshwater lens or exfiltration rate compared to the permeability and geometry of the sedimentary basin. We therefore select the shallow ice model as it is the simplest ice sheet model (having a single parameter, the dimensionless accumulation) that can be solved subject to a flotation condition at the grounding line and no surface slope at the origin. While other approximate models (notably a 'shallow-shelf approximation 'with large basal slip) are possible, they really ought to be coupled back to subglacial hydrology, and that complication is beyond the scope of this study.

With this said, we agree that the manuscript would benefit from a fuller discussion of the use of this model. We have revised the manuscript to include a discussion of this rationale, by including at page 8, line 185
"... rate. **We use this model because it is very straightforward to solve and introduces minimal additional physics to the model.** We also assume...".
We have also added at page 26, line 514
"... subglacial hydrology. **Such a model would also permit the use of a more sophisticated ice sheet model than the shallow ice approximation used in this paper, such as a shallow shelf model (e.g. Morland (1987)) with a basal sliding law coupled to subglacial hydrology. However, the results of Sect. 5 indicate that, when periodic grounding line movement prevents groundwater dynamics from reaching a steady state, the precise shape of the ice sheet has little effect on groundwater flow compared to the permeability and geometry of the sedimentary basin. The resulting model uncertainty is therefore small compared to that resulting from e.g. aquifer heterogeneity, or lateral variations in basement geometry."**

*"The main goal is to provide a long-term perspective of freshwater lenses and trapped subglacial seawater. However, the exclusion of vertical pressure gradients is a significant limitation because past work have shown groundwater flows in Antarctica to be quite sensitive to those. The manuscript includes a discussion with references to the inferred hydrological budget of ice streams at the Siple Coast, but previous work has also modelled the vertical exchange. To give an example, Christoffersen and Tulaczyk (Annals of Glaciology, 2003) included glacial-interglacial simulations of groundwater exchange at the Siple Coast. There may be a relevant discussion in that thermally driven exfiltration is shallow compared to the horizontally driven exchange presented in this manuscript."*

While our manuscript does discuss the neglect of non-hydrostatic vertical pressure gradients and consequent exfiltration arising due to the dynamic response of sediments to loading, we have not considered those due to basal freeze-on. However, we may justify their neglect on the grounds that these vertical pressure gradients typically exist only within the upper few tens of metres of till. Therefore, while this exfiltration is important for calculating basal water budgets, it is less important when modelling the dynamics of groundwater throughout the full depth of a sedimentary basin, which is the focus of our study.

We have added a discussion of this at page 26, line 544
"... the same k. **We have also neglected basal freeze-on, which could drive substantial exfiltration of water from sediments over the timescales of glacial advance and retreat, considered for instance by Christoffersen and Tulaczyk (2003b). However, the dynamic effects of this exfiltration are typically confined to the upper 5–50 m of sediment, meaning that the corresponding pressure gradients do not substantially affect the overall flow of groundwater throughout the depth of the ~1 km thick sedimentary basin. Such exfiltration could be considered in a possible extension of this model that includes heat transport, as discussed above.**"

*"A final couple of questions. Why not use reconstructed air temperature and precipitation records from Antarctic ice cores instead of a periodic function? Presumably, this would provide more direct evaluation of glacial-interglacial changes. "*

As with the use of the shallow ice approximation, we have chosen a mathematically simple description of historic grounding line position. We have done so on the basis that a more complex model introduces so many other sources of uncertainty (e.g. converting from air temperature to grounding-line position) that we think it makes for a cleaner experiment to simply prescribe the evolution of the groundling line position. For the purposes of this paper, it is most important to accurately capture the approximate position and timing of the grounding line minimum, since this determines where and when the freshwater lens must displace intruded seawater.

While it would be possible to obtain a grounding line position by forcing an ice sheet model using real-world ice core data, this requires introducing additional physics (e.g. in modelling the ice flux across the grounding line) and may need heavy fine-tuning to

recover the best existing estimates of the grounding line minimum. In addition, a numerical model forced using the existing ice core record up to the present day may exhibit dependence on the arbitrary initial condition imposed, whereas a periodic forcing ensures the existence of a periodic solution.

We have made a revision on page 19, lines 405-410:
"We choose to use **these** simplified model**s** on the basis that the exact form**s** of $p_S$ **and** $x_g$ **are** a relatively small source of model uncertainty compared to e.g. lateral variations in H and b, or heterogeneity in aquifer properties such as $k$ and $\phi$. **We have therefore prioritised capturing appropriate values for the grounding line maximum and minimum, and their timing. A more sophisticated approach could use ice core data for historic accumulation and temperature to force a model of both $p_S$ and $x_g$, but such a model may require excessive fine-tuning to reproduce existing estimates of the grounding line extrema.**"

*"Also, how sensitive is the exchange of water at the top of the aquifer to the assumed impermeable basement? What if the basement wasn't impermeable?"*

This is an interesting consideration. Our assumption that the basement is impermeable is based on the measurements of Gustafson et al. (2022), which show a significant increase in electrical resistivity below a certain depth, indicating that very little if any groundwater resides in the basement rock. Although this measurement is localised, Tankersley et al. (2022) likewise assume that the permeability of the basement is very low, and that this low permeability is key in confining groundwater flow.
It is an interesting possibility to consider a model in which a different boundary condition, such as a nonzero flux of water, is imposed at the basement. However, such a condition would be somewhat arbitrary in nature as little is known about the basement rock beyond its upper extent and low permeability. If the permeability of the basement rock were significant on the timescales of our consideration, the modelled exfiltration during basin shallowing and infiltration during deepening would be weakened, since some outflux would occur into the basement. However, assuming that the permeability of the basement is low, this effect would be small, and further mitigated if a decrease in basin permeability with depth due to sediment compaction were factored in.
The position of the basement is important for the results of the model, and it is worth noting that this is itself subject to uncertainty. This includes model uncertainty in the inversion of magnetic data by Tankersley et al. (2022), but also the uncertainty introduced by our taking a two-dimensional cross section of this data.

We have made a revision to discuss this at page 25, line 519, following on from the revision mentioned above:
"... subglacial hydrology. **Such a model ... basement geometry.**
**Since basement geometry is important for the results of the model, particularly $q_E$, it should be noted that the use of a cross-sectional model introduces uncertainty even after transverse averaging. Moreover, the data of Tankersley et al. (2022) includes some model uncertainty introduced when inverting magnetic measurements. Future work is therefore required to explore the dynamics of**

**subglacial groundwater flow in all 3 dimensions, which we have discussed in Appendix C. We have assumed, following Gustafson et al. 2022 and Tankersley et al. 2022, that the basement may be treated as impermeable, although an extension of this model could include a basement which is weakly permeable. The inclusion of basement permeability would weaken the effects of basement geometry on $q_E$ by providing an alternative route for groundwater to leave or re-enter the sedimentary basin."**

Reviewer 2:

*I appreciate the layer of water between the ice and the aquifer is thin, but a discussion about typical length scales and modelling assumptions would be useful. To aid discussion, it would perhaps be useful to add a new symbol to denote the underside of the ice, as it is also being referred to in the schematic of Fig. 1. That is, the caption refers to the layer of water between the ice (at z=?) and the aquifer (at z=S).*

The thin layer of water is representative of a 'shallow hydrological system 'including (e.g.) subglacial channels, lakes and till, whose depth is altogether on the order of metres to tens of metres. In contrast, the thickness of the sedimentary basin ranges from hundreds to thousands of metres. We therefore make a modelling assumption that the shallow layer of water has zero thickness. For this reason, we think that adding a new symbol to denote the underside of the ice would be more confusing than helpful, as the model makes no real distinction between this and the upper surface of the aquifer. We have added a sentence to better explain this at line 64:
"... hydrological system. **We treat this layer in the model as having zero thickness, on the basis that the corresponding 'shallow 'hydrological system has a depth on the order of metres to tens of metres, whereas the aquifer is hundreds to thousands of metres deep (Gustafson et al., 2022)."**

*Although it is part of the definition of an aquifer, I would suggest to add "permeable" before 'aquifer" on line 53, for clarity.*

We have implemented this suggestion.

*I understand ice is assumed to deform but the sediment is not. Can something more be said about when such an approximation is valid?*

We regard the aquifer as rigid sedimentary rock, and assume that any sediment deformation is confined to the shallower layer above (discussed above) where it could facilitate 'sliding'. We believe that this is a sensible assumption at first order on the timescales of our consideration. This assumption does not extend to the till comprising the upper few metres of the sedimentary basin, which we treat separately as part of the 'shallow 'hydrological system, and which is likely to deform substantially in response to ice and groundwater dynamics. We have added a sentence at line 73 to clarify the latter:

"exfiltration $q_E(x,t)$. **This assumption of rigidity does not extend to the deformable till comprising the upper few metres of the sedimentary basin, which we regard as part of the basal hydrological system.**"

*I understand that the frequently used sharp-interface approximation, following other works, has been applied. Can a comment be made about how sharp the boundary between the seawater and the aquifer ahead of the grounding line is in practice?*

The salinity profiles obtained by Gustafson et al. (2022) suggest that, in reality, the salinity varies smoothly from fresh to saline throughout the aquifer rather than displaying a distinct sharp interface. Our conclusions involve discussion of the limitations of the sharp-interface assumption, and possible approaches to accounting for mixing of saltwater and freshwater (lines 550–558). We have made a revision to refer to this discussion earlier on at line 77:
"is negligible (Bear, 2013; Mondal et al., 2019). **Since the measurements of Gustafson et al. (2022) suggest a smooth variation in salinity through the aquifer rather than a distinct sharp interface, we include in Sect. 6 a discussion of how a similar model could account for these mixing processes. For now, however, we proceed with the sharp-interface assumption as a useful and tractable first-order model.** The aquifer therefore..."

*How has (5) been obtained? The +/- subscripts are hard to follow.*

Equation (5) states that the velocity of the interface is equal to the Darcy velocity of the water on either side thereof, ensuring that mass is conserved across the interface. We have made a revision to better explain this at line 100:
"... where the subscripts ± **denote the Darcy flux components on either side of the interface, which are not continuous in general.** However, the pressure $p(x,z,t)$ must be continuous. **Equation (5) ensures that the velocity of the interface equals the Darcy velocity of the fluid on either side, so that mass is conserved across the interface.**"

*The switch between dimensional and dimensionless variables is hard to follow. Presumably the equations are in dimensionless form but the figures are dimensional and still use the same notation for all the variables?*

This is correct: from Sect 2.2 onwards, variables are dimensionless unless otherwise stated, but figures are dimensional. We make this choice for readability and ease of interpretation, rather than introducing additional notation to distinguish dimensionless and dimensional variables. We have added a sentence to explain this at line 151:
"**From here onwards, variables are dimensionless unless otherwise stated, but dimensional values are used in figures, based on the scalings provided in Tables 1 and 2 unless otherwise stated.** Under these..."

*I was surprised that the depth of the aquifer is of a similar vertical length scale to ice depth. This would be worth pointing out specifically, perhaps when the nondimensionalisation is made.*

We agree that this reuse of the vertical length scale is worth pointing out, and have made a revision to do so at line 148:

"...and [$z$]. **Since both the aquifer and ice sheet are is hundreds to thousands of metres deep, we may use the latter for both the aquifer and ice sheet.** We also assume..."

*Fig 2: What about the streamlines within the freshwater-saturated aquifer is more interesting than those in the saltwater-saturated aquifer?*

In the steady state, there is no flow in the saltwater-saturated region, and hence there are no streamlines to be plotted in this region. We have added a sentence stating this explicitly at line 231:

"... steady state. **There is no flow in the saltwater region in the steady state.**"

*Fig 3: Are the steady state solutions those obtained with x_g(t)=\bar x_g? I can infer that from line 248, but it would be clearer to state this explicitly.*

This is correct. We have added a sentence to clarify this in the caption of Figure 3:

"... (low permeability). **Instantaneous steady states are plotted for $x_g$= \bar x_g, \bar x_g ± △ x_g.** Parameters are..."

*How can saltwater get to a saltwater pocket in steady state? Does this follow some kind of time-dependent forcing of the grounding line? This isn't made clear at the beginning of Sec. 4, only becoming clearer later on.*

Generally speaking, in order for a saltwater pocket to form, the groundwater flow must evolve from an 'initial condition' where saltwater is present sufficiently far into the aquifer to become trapped in the pocket, such as the examples provided in Figure 5. This is not unreasonable, as without an ice sheet the entire aquifer would be saltwater-saturated. However, in practice, there is always a grounding line history which determines this 'initial condition'; moreover, the 'initial condition' and subsequent pocket formation are highly sensitive to this grounding line history, as seen in Figure 6.

We have added a brief explanation of this at line 288:

"...Figure 4. **Such a pocket may be formed, for example, if the ice has advanced sufficiently rapidly to its current position that saltwater is still present relatively far inland once the ice has halted (see later Figures 5 and 6).** These pockets..."

*Figure 4: A label describing the basal geometry (in the legend) would be useful.*

We have added a description of the basal geometry in the caption on Figure 4:

"...illustrated. **The geometry is given by $S$ = -1000 m, $b$ = ( -2500 + 1000 exp( -( ($x$- 125 km) / 12.5 km )$^2$ ) ) m.** (b) Freshwater..."

*Figure 8: I found it hard to interpret the two black curves as they're unlabelled (why is it two, not one, for examples). It also took some digging to understand where the basement geometry is coming from. Perhaps state the location explicitly here.*

The black curves are the upper ($z=S$) and lower ($z=b$) surface of the aquifer, which is consistent with previous figures. These are estimated by taking an averaged cross-section along a historic flowline of the ice, as described in the previous Figure 7. We have modified the caption of Figure 8 to make this clearer:

"Potential steady states for the basement geometry **$z=b$ and $z=S$ (black curves) estimated along a historic flowline of the ice in the Ross Sea, West Antarctica, as shown in Figure 7. The freshwater-saltwater interface $z=s$ and ice surface are plotted** for various values of the ice sheet parameter $\alpha$."

- *Line 18: Add commas before and after "which drain much of West Antarctica".*
- *1 Caption, final line: Change "z= S" to "z= S(x)".*
- *2 and 4: It is technically incorrect to refer to the cyan line as S(x)+H_i(x) (there's an ice shelf there too), so I suggest to just refer to it as the ice surface, which would be consistent with Figs. 3 and 5.*

We have implemented all of these suggested changes.

**Comment by Giacomo Medici**

*"Line 6. "Two-dimensional groundwater flow". Add text in the discussion section on assumptions and limitations underneath the choice of a 2D model."*
We have added a discussion of the limitations introduced due to the choice of a two-dimensional model as part of our responses to reviewer comments above. The chief source of uncertainty is the inability of a two-dimensional model to account for lateral variations in sedimentary basin geometry. This discussion is found at lines 513–523 of the marked-up manuscript.

*"Lines 30-34. Add relevant and recent literature on tracer and hydraulic tests in sedimentary deposits of glacial origin made by clay, sand, breccias and conglomerates: - Tracking flowpaths in a complex karst system through tracer test and hydrogeochemical monitoring: Implications for groundwater protection (Gran Sasso, Italy). Heliyon, 10(2). - Forms of hydraulic fractures created during a field test in overconsolidated glacial drift. Quarterly Journal of Engineering Geology and Hydrogeology, 28(1), 23-35."*

We are aware that there is extensive literature on the various methods used to investigate groundwater flow and hydraulic properties of aquifers, of which the suggested papers are a good example. However, we have limited the focus of our study to groundwater flow beneath current marine ice sheets, so that the aquifer is either covered by ice or ocean at all times. Aquifers which are exposed to the air (e.g. following retreat of a land-terminating ice sheet) are subject to different physics (for example, recharge is determined by precipitation rates rather than the imposed pressure at the ice bed). We therefore exclude such aquifers, which comprise the majority of well-studied glacial deposits including those referred to in the suggested papers, from our study. As a result, we do not believe that the suggested literature is directly relevant to the focus of our paper, although it would be interesting to incorporate such research into future generalisations of this model to a broader class of settings.

[Add]

*"Line 510. "Complex model" to develop in the future. Do you mean a model with multiple units to account for the heterogeneities of the system?*
*Line 510. "Complex model" do you also mean more attention on the anisotropies? You mention heterogeneities in the manuscript, but not anisotropies"*

In this sentence a "more complex model" refers to separately modelling the dynamics of the shallow hydrological system. We have changed the wording at line 510 to help to clarify this:
"A more complex model **that relates** $q_E$ and $p_e$ by **separately** accounting for the dynamics of the shallow hydrological layer, and **that considers** both freshwater and saltwater, would provide ..."

We have also made changes to include discussion of anisotropy (as distinct from heterogeneity) in the permeability *k* as a source of uncertainty at the following:
Line 413: "This includes model uncertainty (e.g. possible spatial heterogeneity in $\phi$ and k, **dependence of k on $\phi$, and possible anisotropy in *k*)**"
Line 484: "Our chosen value of *k* represents an idealisation of a sedimentary aquifer which may in reality be highly heterogeneous **and anisotropic."**
Line 518: "The resulting model uncertainty is therefore small compared to that resulting from e.g. aquifer heterogeneity **and anisotropy**..."
Line 627: "When the porosity $\phi$ (x,z) and permeability k(x,z) are heterogeneous **(but k remains isotropic)**,..."

*"Lines 655-755. Add the recent literature suggested above on the glacial environment."*
As discussed above, we do not believe that the suggested literature is directly relevant to the strict focus of our study.

*"Figure 1. Do you need an approximate spatial scale for your conceptual model?"*
Since the horizontal and vertical lengthscales are discussed elsewhere (e.g. at line 66, or later at Table 1), we think that adding a spatial scale to Figure 1 would be unnecessary and result in a more cluttered Figure. We have therefore opted to keep this Figure unchanged.

*"Figure 3. Very busy figure, consider to split it in two parts."*
After consideration we believe that the current format is best, as it is helpful to see advance and retreat together to appreciate the periodicity of the solution. It is also helpful to directly compare the solutions for different K, as the main purpose of the Figure is to illustrate the difference and resulting saltwater trapping. Finally, depicting $q_E$ directly below the timeseries of s(x,t) gives some idea of how the flow behaves, and is consistent with the previous Figure 2 and following Figure 4.

*"Figure 6. There is room to make the figure larger."*
*"Figure 11. Same here, there is room to make the figure larger. The figure would benefit from that."*

We have made both of these figures larger in response to this feedback.

**Comment by Matthew Tankersley:**

*"I noticed a few small errors in the citation of my work which I thought I would inform you of. The sedimentary basins we discussed were modeled with airborne magnetics data, not imaged with radar data as mentioned in the text. I think this is an important distinction as the modeling aspect, as opposed to direct imaging, introduces a lot more uncertainty which your readers should be aware of, and magnetic and radar techniques are quite different. No worries, but if you're able to change your mentions of radar to magnetics that would be great."*

We have corrected references to radar measurement in the context of the paper by Tankersley et al. (2022) to refer instead to modelling using airborne magnetic data. These are found at lines 22, 378, and 385 of the marked up-manuscript.

**Other corrections:**

A typo in Equation (19) (an incorrect leading minus sign) has been removed.

Grammar has been improved at lines 286 and 560, and a typo has been corrected at line 303.

---

## Referee Report (RR1)

**Review: "Groundwater dynamics beneath a marine ice sheet"**

by Cairns et al.

Submitted to *The Cryosphere*

**1 General**

In this paper, the authors analyze the flow of water in the porous till below ice sheets. This is a valuable paper with insightful calculations. At the same time, the paper is a tome, aiming to accomplish many goals with several ideas. From what I can tell, the paper has gone through a round of revisions already before it has come to my desk, so my comments are brief and focused on how to improve the paper.

**2 Specific comments**

1. Abstract: the last sentence is ambiguous and could be clarified. Also, it could be beneficial to zoom out and briefly state the significance of results.

2. Smith et al. (2020) is not a great reference for motivating contributions to future sea-level rise. What about a paper like Seroussi et al. (2020)? Or both?

3. line 20: could add 'potential' between 'important' and 'contributor', to reduce the certainty of the statement to a level comparable with the evidence.

4. line 100: could define effective pressure. It is implicit, but could be clarified.

5. line 140: it could be valuable to explain a bit more about which grounding line position and aquifer thickness are good scales. Is it the initial value? Could be more clearly stated.

6. The nondimensionalization is a little hurried. I think specifying clearly the valuables that are scaled and by what would be valuable. This is clearly needed since the first equations after the nondimensionalization have $h$ and $H$ in them. This is confusing if you just scaled the height $H$ by $H$.

7. Section 2.3: with zero effective pressure and a focus on ice streams, it is hard to imagine that the shallow-ice approximation is the right limit of the Stokes equations. There will likely be more than 'negligible bed slip'. At this stage in the review process, the best I can hope for is a clearer description of why this model was chosen, the drawbacks, and later in the paper, how it affects your results.

8. figure 2: does the solution become singular at $x = 0$?

9. paragraph at line 270: I think the relationship between $q_E$ and $K$ could be clarified with a figure.

10. paragraph at line 360: it could be valuable, given the venue at *The Cryosphere*, to describe some of the implications of the hysteresis.

11. section 5: I think this part of the paper could be its own paper. That would allow for more discussion of the results in all sections. Currently, the text continues to be hurried.

12. How does this model compare to the SLW salinity measurement? It seems like text would be devoted to this point – did I miss it?

13. I like the conclusions section, it is a nice wrap up of the paper, much like an expanded discussion section. The paper would benefit from more discussion generally.

**References**

H. Seroussi, S. Nowicki, A. J. Payne, H. Goelzer, W. H. Lipscomb, A. Abe-Ouchi, C. Agosta, T. Albrecht, X. Asay-Davis, A. Barthel, R. Calov, R. Cullather, C. Dumas, B. K. Galton-Fenzi, R. Gladstone, N. R. Golledge, J. M. Gregory, R. Greve, T. Hattermann, M. J. Hoffman, A. Humbert, P. Huybrechts, N. C. Jourdain, T. Kleiner, E. Larour, G. R. Leguy, D. P. Lowry, C. M. Little, M. Morlighem, F. Pattyn, T. Pelle, S. F. Price, A. Quiquet, R. Reese, N.-J. Schlegel, A. Shepherd, E. Simon, R. S. Smith, F. Straneo, S. Sun, L. D. Trusel, J. Van Breedam, R. S. W. van de Wal, R. Winkelmann, C. Zhao, T. Zhang, and T. Zwinger. ISMIP6 Antarctica: a multi-model ensemble of the Antarctic ice sheet evolution over the 21st century. *Cryosphere*, 14(9):3033–3070, 2020. doi: 10.5194/tc-14-3033-2020.

B. Smith, H. A. Fricker, A. S. Gardner, B. Medley, J. Nilsson, F. S. Paolo, N. Holschuh, S. Adusumilli, K. Brunt, B. Csatho, K. Harbeck, T. Markus, T. Neumann, M. R. Siegfried, and H. J. Zwally. Pervasive ice sheet mass loss reflects competing ocean and atmosphere processes. *Science*, 368(6496):1239–1242, 2020. doi: 10.1126/science. aaz5845.

---

## Author Response (AR2)

Dear Dr McCormack,

Thank you for inviting us to respond to these reviews. Please find below our point-by-point response to the reviewer comments along with the corresponding revisions made to the manuscript. In response to the Reviewers we have made changes to clarify our explanation of the method, and our reasoning behind certain modelling choices (in particular the sharp-interface assumption, and our choice of ice sheet model). We have also sought to add further discussion of results and model limitations, or make existing discussion clearer, where the reviewers have indicated a need for this. Line numbers where used, refer to the draft manuscript seen by the reviewers.

Best regards,
Gabriel Cairns

Reviewer 1

*Note: I was not a reviewer for the first round of comments, so I don't have detailed feedback on whether the first round of suggestions were implemented to those reviewers' satisfaction. However, regarding the sharp interface assumption, I do think this point could be discussed a bit more thoroughly, e.g. giving the Peclet number for this system. This is a moment where the model disagrees with the available observations; it would be nice to add some heuristic discussion (if possible) about how adding mixing might alter the conclusions. (I realise it is far too much to implement in this work, but in what direction are the effects more likely to drive the qualitative results?)*

We have expanded our discussion of the sharp-interface assumption at lines 75–80 to more thoroughly justify the use of this assumption, with reference to the estimated size of the Peclet number.

"We use the sharp-interface approximation, a frequent assumption in saltwater intrusion problems, which asserts that mixing between freshwater and saltwater through molecular diffusion and hydrodynamic dispersion is negligible (Bear, 2013; Mondal et al., 2019). Since the inversion results of Gustafson et al. (2022) suggest a smooth variation in salinity through the aquifer rather than a distinct sharp interface, we include in Sect. 6 a discussion of how a similar model could account for these mixing processes, **although it is possible that this smoothing could be in part due to the inversion method used.**
**The sharp-interface approximation is warranted provided that the Péclet numbers associated with molecular diffusion and hydrodynamic dispersion are large (Bear, 2013; Dentz et al., 2006; Koussis and Mazi, 2018). For a molecular diffusivity $D$=10$^{-9}$ m$^2$ s$^{-1}$ and a hydrodynamic dispersivity length $A$=10$^{-2}$ m, which are appropriate for salt in groundwater in sedimentary rocks (Aquilina et al. (2015)), these Péclet numbers are respectively**
$$Pe_{diff} = [z]^2 / D [t] \approx 300, \qquad Pe_{disp} = [z]^2 / A[x] \approx 200,$$
**suggesting that the sharp-interface assumption is reasonable here."**

We have also modified the paragraph at 551–559, to expand our discussion of how including mixing in the model might affect the conclusions:
"Our model has made the assumption of a sharp interface dividing freshwater and saltwater, **on the basis that the Péclet numbers associated with diffusion and hydrodynamic dispersion are large**. This greatly simplifies the task of solving the problem, at the cost of being unable to account for the smoothly varying salinity modelled by Gustafson et al. (2022), **although this smoothness may in part a result of the model used to invert the magnetotelluric data.** To account for the effect of mixing, it is possible (though more complicated) to solve the full problem of density-coupled salt transport, known as the Henry problem, (e.g. Croucher and O'Sullivan (1995)) or to introduce a "mixing layer" around the sharp interface where these effects are accounted for (Van Duijn and Peletier, 1992; Paster and Dagan, 2007). **In steady-state seawater intrusion problems, the inclusion of saltwater-freshwater mixing reduces the extent of seawater intrusion by inducing a circulatory flow in the saltwater-saturated region (Cooper (1959); Koussis and Mazi (2018)). We therefore expect that including mixing would result in a more retreated "interface" (which would need to be newly defined as e.g. the 50% salinity contour) than that achieved with the sharp-interface assumption at the same permeability. This would lead to a lower permeability estimate than that obtained above. The process of saltwater mixing would also reduce the trapping of saltwater in pockets via entrainment into the freshwater flow."**

*On a related note, the conclusions have become quite a sprawling discussion section, so could perhaps be formally split up - it's a little jarring to be directed to the conclusions when considering model limitations in section 2.*

To distinguish discussion of e.g. the limitations of the model and possible extensions from our conclusions, we have split up the "Conclusions" section so that lines 489–559 are now referred to as "Discussion", and the "Conclusions" section consists of lines 560–569.

*I also wondered about the effects of residual trapping at a pore scale, rather than an aquifer-wide one; I appreciate it is too late in the review process to ask for much more than a comment here.*

We do not expect pore-scale residual trapping to be relevant to this problem, because residual trapping occurs as a result of the surface tension between two immiscible fluids, whereas saltwater and freshwater are miscible and thus have no surface tension between them. We have added a comment at line 80 following on from our changes to the discussion above:
"...**reasonable here. Although we assume a sharp interface between the two regions, the fluids are miscible, meaning that there is no surface tension between them. We can therefore ignore the possibility of residual trapping at the pore scale (Bear, 2013).**
The aquifer therefore..."

*Minor comments:*

*Line 273: An example with K = 0.01, while perhaps not that different from K = 0.1, would help illustrate the proposed convergence of the solutions as K goes to 0.*

An example with *K*=0.01 is shown in the following figure (as in Figure 3, the columns correspond to advance and retreat respectively):

[Figure]

This example supports our hypothesis that the saltwater-freshwater interface approaches the quasi-steady state for the most retreated grounding line position as K tends to 0.

Figure 3 is already large and fairly dense with information, and we do not believe that adding the above example with *K*=0.01 would add much to the figure which is not already seen in the *K*=0.1 example. Replacing the latter with the former is also undesirable because the different interface positions are easier to distinguish in the *K*=0.1 example. We therefore propose to include the above figure in the Supplementary Materials, with reference in the caption of Figure 3:

"...For full solutions see Supplementary Animations S1–S3. **An example for K=0.01 is shown in Supplementary Figure S1.**"

as well as in the main text at line 275

"...position. **(For an example with K = 0.01, which supports this hypothesis, see Supplementary Figure S1).**"

*Line 275: Could you clarify this comment about "finite time" - at least at the surface of the aquifer, there is an instantaneous freshening/salting as the grounding line moves back and forth. Do you mean at depth? But then the seawater intrusion isn't instantaneous at depth either.*

By "finite time", we mean that, during grounding line advance, the freshwater lens grows at a finite rate, determined by the flux of freshwater $-q_E$ into the aquifer from above. Therefore, it takes finite time for freshwater to penetrate to any given depth below the surface. In contrast, during grounding line retreat, the saturation of the region $x > x_g$ ahead of the grounding line throughout its depth is instantaneous. We have made a change of wording at line 275 to clarify this, following on from the comment added above:

"**... see Supplementary Figure S1). This is because, during grounding line retreat, seawater instantaneously displaces any freshwater from the newly exposed region**

**$x>x_g$. However, during re-advance, the freshwater lens grows at a finite rate, determined by the infiltration $q_E<0$, and depending on $K$.** This is the key…"

*Line 365: Why was K = 0.5 chosen as an example given that previous illustrations took powers of 10?*

The value $K=0.5$ was selected because it is one of the higher values of $K$ which leads to an eventual 'pocket' steady state in the example of Figure 6. A higher value of $K$ is favourable in this instance because it enables quicker relaxation to the steady state, allowing the final states to be compared as seen in Figure 6(d). Choosing e.g. $K=0.1$ would produce the same final state but would also require a longer relaxation time to do so, which would affect the readability of Figures 6(e) and 6(f). We have added a sentence to explain this at line 365:

"…parameter $K$. **In this case we select values of $K$ that illustrate a difference in final states whilst being relatively fast to approach these states, for the sake of readability of the figure.**"

*Line 376: Perhaps "more realistic" rather than "real-world" given the number of model simplifications*

We agree that this wording is more appropriate and have made the suggested change.

*Some floating "s at lines 527, 550, no closed bracket line 573*

Thank you for pointing these errors out. We have corrected them.

Reviewer 2

*1. Abstract: the last sentence is ambiguous and could be clarified. Also, it could be beneficial to zoom out and briefly state the significance of results.*

We have decided to remove the last sentence of the abstract (at line 12) due to its ambiguity. We have also added a sentence at the end of the abstract discussing the significance of our results:
"…becomes deeper. **Our results highlight the potential importance of groundwater flow in sedimentary basins for the subglacial hydrology of ice streams.**"

*2. Smith et al. (2020) is not a great reference for motivating contributions to future sea-level rise. What about a paper like Seroussi et al. (2020)? Or both?*

We agree that when considering future contributions to sea level rise, a paper such as Seroussi et al. (2020) modelling future evolution of the Antarctic ice sheet is a better reference than the observations of recent historic sea level contributions by Smith et al. 2020. We have therefore changed the reference at line 15 accordingly.

*3. line 20: could add 'potential' between 'important' and 'contributor', to reduce the certainty of the statement to a level comparable with the evidence.*

We have made the suggested change of wording to reflect this.

*4. line 100: could define effective pressure. It is implicit, but could be clarified.*

We have modified the sentence at line 100 to clarify the definition of effective pressure: "...is equivalent to stipulating that the "effective pressure", **defined as $p_e = \rho_i g H_i - p$**, is zero."

*5. line 140: it could be valuable to explain a bit more about which grounding line position and aquifer thickness are good scales. Is it the initial value? Could be more clearly stated.*

We have added additional wording at line 147 to clarify the basis for the choice of scales:
"We assume that **the present-day grounding line position, measured relative to the onset of the sedimentary basin (Peters et al., 2006), and the approximate aquifer thickness at the measurement sites of Gustafson et al., 2022,** provide suitable horizontal and vertical lengthscales *[x]* and *[z]*."

*6. The nondimensionalization is a little hurried. I think specifying clearly the variables that are scaled and by what would be valuable. This is clearly needed since the first equations after the nondimensionalization have h and H in them. This is confusing if you just scaled the height H by H.*

We have added more detail to our explanation of the non-dimensionalisation process, to clarify our use of notation and make clearer the variables that have been scaled and what they have been scaled with. This is found at line 151:
"We then scale
$$x, x_g \sim [x], \qquad z, h, H, b, s, S \sim [z], \qquad t \sim [t], \qquad p_S \sim \rho_f g[z], \qquad q_E \sim [z]/[t]. \qquad (21)$$
**For example, we let $x = [x] x\hat{\,}$, where $x\hat{\,}$ is non-dimensional, and do likewise for each of the variables listed with a corresponding scale in Equation (21) (i.e. $z = [z] z\hat{\,}$, $h = [z] h\hat{\,}$ etc.). We then work with these non-dimensional variables but drop all hats $\hat{\,}$ from the notation.** From here onwards..."

*7. Section 2.3: with zero effective pressure and a focus on ice streams, it is hard to imagine that the shallow-ice approximation is the right limit of the Stokes equations. There will likely be more than 'negligible bed slip'. At this stage in the review process, the best I can hope for is a clearer description of why this model was chosen, the drawbacks, and later in the paper, how it affects your results.*

We agree that a version of the shallow-shelf approximation (SSA) provides a more appropriate model for the glaciological context, without affecting the simplicity of the ice sheet model. Specifically, using the SSA combined with a Weertman sliding law, neglecting extensional stresses, and assuming a prescribed grounding line position,

results in a similar equation to that obtained under the SIA. We have rerun our code using this SSA model, and updated Figures 2–6 and 8–11 and Table 3 accordingly. The resulting modelled ice thickness is very similar to that previously obtained using the SIA, resulting in a small quantitative change to our results but no significant qualitative change.

Other choices of sliding law are possible, such as a regularised Coulomb sliding law (e.g. Schoof, 2005), but these typically involve some dependence on the effective pressure at the ice bed. Incorporating the effective pressure into the sliding law ultimately requires a model of shallow hydrology, which we consider beyond the scope of this paper. Indeed, our assumption of zero effective pressure is incompatible with several sliding laws (e.g. Budd, regularised Coulomb), which are singular at zero effective pressure.

We have updated our description of the ice sheet model to reflect this change:
"…by Eq. (7). **To model the ice sheet, we use the shallow-shelf approximation (MacAyeal, 1989; Schoof, 2007), assuming a Weertman sliding law with exponent 1/3 and that extensional stresses in the ice are negligible. We also assume that the accumulation is spatially uniform, and that the dynamics of the ice sheet are quasi-steady for a given $x_g$. This leads to equations of mass and momentum conservation**
$$H_i u_i = ax, \qquad \beta u_i^{1/3} = \rho_i g H_i \partial/\partial x \, (H_i + S),$$
**where $u_i$ is the (vertically uniform) ice velocity, $a$ is the constant accumulation rate measured in m s$^{-1}$, and $\beta$ is a constant in the sliding law measured in Pa m$^{-1/3}$ s$^{1/3}$. The latter equation represents a balance between the driving stress and the basal friction, which has a power-law dependence on the velocity under the Weertman sliding law. These together lead to the dimensional equation**
$$(\beta/\rho_i g)^3 H_i^4 \, (\partial/\partial x \, (H_i + S) \,)^3 = ax.$$
**In dimensionless variables, after scaling Hi with [z], Equation (7) and (30) become**
$$p_S = r_i H_i, \qquad H_i^4 \, (\partial/\partial x \, (H_i + S) \,)^3 = \alpha x,$$
**where**
$$r_i = \rho_i / \rho_f \approx 0.917, \qquad \alpha = a\beta^3 [x]^4 / (\rho_i g)^3 [z]^7."$$

Because we have removed the need for the Glen's law parameter $A$, and introduced a new sliding parameter $\beta$, we have also updated the references to $A$ at Table 1 and at line 200 to reflect the new ice sheet model.

We have also added a discussion of why this ice sheet model has been selected at line 201:
"**We have chosen the above ice sheet model because it is simple and quick to solve for a given grounding line position and is physically appropriate for an ice stream context, where fast ice flow is dominated by basal slip (Cuffey and Paterson, 2010). This is because our main purpose is to model the dynamics of groundwater, rather than those of the ice sheet. A more general model could replace the Weertman sliding law with other sliding laws. However, such sliding laws generally involve coupling to the effective pressure in the shallow**

hydrological system, the modelling of which is beyond the scope of this paper. We include in Sect. 6 a discussion of the limitations of this model, and of how it might be generalised."

In the discussion section, at lines 514–518, we have expanded our discussion of how our ice sheet model might affect our results:
"Such a model would also allow for coupling between the ice sheet model and subglacial hydrology. This could be achieved, for instance, by replacing the Weertman sliding law used in the ice sheet model above with a sliding law coupled to the effective pressure $p_e$ (e.g. a regularised Coulomb sliding law (Schoof, 2005)). The use of such an ice sheet model would likely lead to ice that is thinner and shallower near the grounding line, leading to more saltwater intrusion in the steady state and a higher likelihood of trapping seawater in pockets, as well as leading to smaller exfiltration / infiltration. However, the same modelled ice sheet may be steeper inland, resulting in a larger $|q_E|$ further inland. It is also possible that feedbacks could exist between groundwater flow and the ice sheet. For example, groundwater exfiltration could lower the effective pressure in the shallow hydrological system, which would affect the sliding of the ice and hence the profile of the ice sheet, whose gradient feeds back into $q_E$. Since the SLW site is relatively near the grounding line at the present day, we expect that an ice sheet model which predicts shallower ice here would lead to a thinner freshwater lens, and hence a higher predicted permeability.** However, the results of Sect. 5 **suggest** that, when periodic grounding line movement prevents groundwater dynamics from reaching a steady state, the shape of the ice sheet **is less important than** the permeability and geometry of the sedimentary basin."

*8. figure 2: does the solution become singular at x = 0?*

We have added a sentence at line 231:
"...balancing one another in the steady state. **The infiltration $q_E<0$ is large in magnitude but finite at $x=0$, because the ice overpressure gradient $\partial P_S/\partial x$ is zero at $x=0$ whilst changing over a relatively short distance.**"

*9. paragraph at line 270: I think the relationship between qE and K could be clarified with a figure.*

Since the aim of Figures 3(a–c)(iii) and (iv) is to illustrate how $q_E$ is affected by $K$, we do not think that another figure would add much in the way of illustrating this point. We have however made changes to the wording of this paragraph to emphasise the link to these figures and the equation $q_E$:
"The exfiltration flux $q_E$ also depends strongly on $K$, as shown in Figures 3(a)–(c)(ii). **Equation (27) shows that $q_E$ is multiplied through by a factor of $K$, but also depends on the position of the interface s, which is itself affected by $K$.** When $K$ is large **(Figures 3(a)(iii) and (iv)), $q_E$ therefore follows s in being close to its** instantaneous quasi-steady state value, resembling that shown in Figure 2(b)(i). However, when $K$ is small **(Figures 3(c)(iii) and (iv))**, exfiltration **is small apart from** a prominent peak near $x_g$ during grounding line retreat, where freshwater is rapidly flushed out of the aquifer **as**

**seawater saturates the region beyond the grounding line**. Seawater intrusion **could therefore significantly influence the shallow hydrological system via its effect on** $q_E$, as we shall see again in Sect. 5."

*10. paragraph at line 360: it could be valuable, given the venue at The Cryosphere, to describe some of the implications of the hysteresis.*

We have added more discussion of this hysteresis at line 373:
"...possible steady states. **This hysteresis would affect the exfiltration $q_E$ into the shallow hydrological system, because the presence of a saltwater pocket influences $q_E$ (e.g. as seen in Figure 4). This in turn would modify the distribution of water in the shallow hydrological system beneath an ice sheet and could therefore feed back on the dynamics of the ice via basal sliding.**"

*11. section 5: I think this part of the paper could be its own paper. That would allow for more discussion of the results in all sections. Currently, the text continues to be hurried.*

We believe that keeping Section 5 works better when combined with the rest of this paper, since the analysis of the model in idealised scenarios in Sections 3 and 4 helps provide insight into the results of the more complex experiment in Section 5, and the results of Section 5 are still obtained under a number of idealising assumptions that are discussed in the paper. It would be interesting to conduct an experiment similar to that in Section 5 using a more complex model which relaxes a number of these assumptions, which we leave for a future study.
We believe that we have addressed the need for more discussion via our responses to the other reviewer comments.

*12. How does this model compare to the SLW salinity measurement? It seems like text would be devoted to this point – did I miss it?*

The paragraph at lines 443–447 discusses the results for the present-day freshwater lens depth, provided in Table 3, and compares them to the salinity results of Gustafson et al. (2022). We have made changes to this paragraph to improve the discussion of this point:

Table 3 shows the present-day depth of the transient freshwater lens $d_{f,SLW}$ at the location of SLW in the above solutions. This increases with the permeability $k$, reaching the full depth H of the layer for the very high value of $1 \times 10^{-10}$ m$^2$, where the solution is near quasi-steady. **Modelled salinity profiles found by inversion of magnetotelluric data by Gustafson et al. (2022) suggest that the groundwater salinity reaches that of seawater around 600m below the ice bed, and 50% of this salinity about 400m below the ice bed. These profiles show a smooth increase in salinity as a result of either mixing of freshwater and saltwater or due to the inversion process itself, as opposed to our results obtained assuming a sharp freshwater-saltwater interface. This introduces some uncertainty in how best to infer the 'freshwater lens depth' from these results. However, it is clear that the high permeability value k = 3 × 10$^{-12}$**

**m$^2$, for which d$_{f,SLW}$ = 400 m, gives the results that can be considered to agree best with those of Gustafson et al. (2022). We discuss how this model might be adapted to allow smoothly varying salinity in Sect. 6.**

*13. I like the conclusions section, it is a nice wrap up of the paper, much like an expanded discussion section. The paper would benefit from more discussion generally.*

As above, we believe that we have addressed the need for further discussion via our responses to the other reviewer comments.

Reviewer 3 (Lu Li)

*This paper addresses an important problem related to understanding long-term groundwater flow in sedimentary basins beneath marine ice sheets. It is technically sound and represents a valuable contribution to the field. Thanks for this interesting work. I really enjoyed reading it. The manuscript has undergone thorough review by several referees and community members, and the authors have responded effectively to the comments. As a result, the paper has been significantly improved and is now ready for publication. A few minor edits related to the geophysical aspects could be made in the final version.*

*Much of our understanding of the geometry and properties of sedimentary basins (e.g., salinity) comes from geophysical data. It would be useful to distinguish the direct observation and model product. Sedimentary basin thickness can be modeled using magnetic data: the magnetic field is directly observed, while the basin thickness is a model product targeting the magnetic basement (assume sediment is non-magnetic). Similar to the approach used in Gustafson's work, magnetotellurics (MT) does not measure salinity directly. Instead, it measures electric and magnetic fields, which are used to invert for electrical conductivity. Salinity is then estimated from the conductivity results using Archie's empirical law.*

*It is somewhat difficult to determine whether there is a smooth transition from freshwater to saline water, as the inversion method used in Gustafson et al. (2022) is designed to produce smooth models that fit the data. It would be good to know the limitations of using geophysics (especially when it is used to constrain the numerical model!). Here is a review on the use of geophysics to investigate sedimentary basins in Antarctica (see Aitken et al., 2023), which hopes to help bridge the geophysical and modeling communities.*

We have modified our discussion at lines 75–80, 443–447 and 551–559 of the smooth variation in salinity, to include the fact that the observed smoothness may be in part a result of the inversion method used – see response to Reviewers 1 and 2 above.

*Below are several suggestions:*

*Line 22: 'Airborne magnetic measurements' to 'Airborne magnetic investigation'. Maybe*

*also add a sentence in the beginning to lead this paragraph: Geophysical data provide constraints on sedimentary basin geometries and properties (Aitken et al., 2023).*

We have made the suggested changes to introduce these geophysical methods, to refer to the suggested literature, and to more accurately describe the nature of these methods:
**"Geophysical data can provide constraints on the geometry and properties of Antarctic sedimentary basins (Aitken et al., 2023; Li and Aitken, 2024). For example, airborne magnetic investigations** of the sedimentary…"

*Line 77: 'the measurements of Gustafson' to 'the inversion results of Gustafson'*
*Line 386: 'come from airborne magnetic measurements by Tankersley et al. (2022)' to 'come from Tankersley et al. (2022) which is derived from airborne magnetic data'*

We also have made these changes of wording.

*Line 387: We cannot direct measure bathymetry from satellite, the primary source of bathymetry from Bedmap2 is active seismic sounding. Just would like to point out that there is a major update of bathymetry beneath the Ross ice shelf using airborne gravity data (Tinto et al., 2019), that's a better option if you want to some future work, or you can use Bedmap3 (Pritchard et al., 2025). You can just say we use the bedrock topography and bathymetry data for S(x) from Fretwell et al., (2013).*

We have made the suggested change of wording at line 387 to correct this.

*Line 388: It's okay to do interpolate the basement geometry in here. Just want to point out there is a first order of sedimentary basin thickness product available (Li and Aitken, 2024), if some future work would like to do the simulation in other areas in Antarctica.*

We have added reference to this research in the introduction (see above).

*Line 392: 'measured' to 'modelled'*
*Line 552: 'for the smoothly varying salinity observed in the measurements of Gustafson et al. (2022)' to 'for the smoothly varying salinity modelled by Gustafson et al. (2022)'*

We have also made these suggested changes of wording.

Other corrections:

We have corrected the values of $k$, $\varphi$ and $K$ in Tables 1 and 2 to those used to produce results in Section 5. A typo has also been corrected at line 451. Small changes of notation have been made at lines 436 and 444 for consistency.

References:

Aitken, A. R., Li, L., Kulessa, B., Schroeder, D., Jordan, T. A., Whittaker, J. M., ... & Siegert, M. J. (2023). Antarctic sedimentary basins and their influence on ice-sheet dynamics. Reviews of Geophysics, 61(3), e2021RG000767.

Aquilina, L., Vergnaud-Ayraud, V., Les Landes, A. A., Pauwels, H., Davy, P., Pételet-Giraud, E., Labasque, T., Roques, C., Chatton, E., Bour, O., et al. (2015). Impact of climate changes during the last 5 million years on groundwater in basement aquifers, Sci. Rep.-UK, 5, 1–12.

Bear, J.: Dynamics of fluids in porous media, Courier Corporation, 2013
Koussis, A. D., & Mazi, K. (2018). Corrected interface flow model for seawater intrusion in confined aquifers: relations to the dimensionless parameters of variable-density flow. Hydrogeology Journal, 26(8).

Cooper Jr, H. H. (1959). A hypothesis concerning the dynamic balance of fresh water and salt water in a coastal aquifer. Journal of Geophysical Research, 64(4), 461-467.

Cuffey, K. M. and Paterson, W. S. B. (2010). The physics of glaciers, Academic Press.

Gustafson, C. D., Key, K., Siegfried, M. R., Winberry, J. P., Fricker, H. A., Venturelli, R. A., and Michaud, A. B. (2022). A dynamic saline groundwater system mapped beneath an Antarctic ice stream, Science, 376, 640–644.

Li, L., & Aitken, A. R. A. (2024). Crustal heterogeneity of Antarctica signals spatially variable radiogenic heat production. Geophysical Research Letters, 51(2), e2023GL10620

MacAyeal, D. R. (1989). Large-scale ice flow over a viscous basal sediment: Theory and application to ice stream B, Antarctica, Journal of Geophysical Research: Solid Earth, 94, 4071–4087

Muszynski, I. and Birchfield, G. (1987). A coupled marine ice-stream–ice-shelf model, Journal of Glaciology, 33, 3–15

Peters, L. E., Anandakrishnan, S., Alley, R. B., Winberry, J. P., Voigt, D. E., Smith, A. M., & Morse, D. L. (2006). Subglacial sediments as a control on the onset and location of two Siple Coast ice streams, West Antarctica. Journal of Geophysical Research: Solid Earth, 111(B1).
Pritchard, H. D., Fretwell, P. T., Fremand, A. C., Bodart, J. A., Kirkham, J. D., Aitken, A., ... & Zirizzotti, A. (2025). Bedmap3 updated ice bed, surface and thickness gridded datasets for Antarctica. Scientific data, 12(1), 414.

Seroussi, H., Nowicki, S., Payne, A. J., Goelzer, H., Lipscomb, W. H., Abe Ouchi, A., Agosta, C., Albrecht, T., Asay-Davis, X., Barthel, A., et al. (2020). ISMIP6 Antarctica: a multi-model ensemble of the Antarctic ice sheet evolution over the 21 st century, The Cryosphere Discussions, 2020, 1–54

*Schoof, C. (2005). The effect of cavitation on glacier sliding, Proceedings of the Royal Society A: Mathematical, Physical and Engineering Sciences, 461, 609–627*

*Schoof, C. (2007). Ice sheet grounding line dynamics: Steady states, stability, and hysteresis, J. Geophys. Res.-Earth, 112*

*Smith, B., Fricker, H. A., Gardner, A. S., Medley, B., Nilsson, J., Paolo, F. S., ... & Zwally, H. J. (2020). Pervasive ice sheet mass loss reflects competing ocean and atmosphere processes. Science, 368(6496), 1239-1242*

*Tinto, K. J., Padman, L., Siddoway, C. S., Springer, S. R., Fricker, H. A., Das, I., ... & Bell, R. E. (2019). Ross Ice Shelf response to climate driven by the tectonic imprint on seafloor bathymetry. Nature Geoscience, 12(6), 441-449.*